# Bridging Arbitrary and Tree Metrics via Differentiable Gromov Hyperbolicity

**Pierre Houedry**
Université Bretagne Sud
IRISA, UMR 6074, CNRS
pierre.houedry@irisa.fr

**Nicolas Courty**
Université Bretagne Sud
IRISA, UMR 6074, CNRS
courty@univ-ubs.fr

**Florestan Martin-Baillon**
Université de Rennes
IRMAR, UMR 6625, CNRS
florestan.martin-baillon@univ-rennes.fr

**Laetitia Chapel**
L'Institut Agro Rennes-Angers
IRISA, UMR 6074, CNRS
laetitia.chapel@irisa.fr

**Titouan Vayer**
INRIA, ENS de Lyon, CNRS, Université Claude Bernard Lyon 1
LIP, UMR 5668
titouan.vayer@inria.fr

## Abstract

Trees and the associated shortest-path tree metrics provide a powerful framework for representing hierarchical and combinatorial structures in data. Given an arbitrary metric space, its deviation from a tree metric can be quantified by Gromov's $\delta$-hyperbolicity. Nonetheless, designing algorithms that bridge an arbitrary metric to its closest tree metric is still a vivid subject of interest, as most common approaches are either heuristical and lack guarantees, or perform moderately well. In this work, we introduce a novel differentiable optimization framework, coined DELTAZERO, that solves this problem. Our method leverages a smooth surrogate for Gromov's $\delta$-hyperbolicity which enables a gradient-based optimization, with a tractable complexity. The corresponding optimization procedure is derived from a problem with better worst case guarantees than existing bounds, and is justified statistically. Experiments on synthetic and real-world datasets demonstrate that our method consistently achieves state-of-the-art distortion.

## 1 Introduction

The notion of $\delta$-hyperbolic metric spaces, introduced by [21], captures a large-scale analog of negative curvature and has found significant applications in both mathematics and computer science. A prominent example in machine learning is the success of representation learning in hyperbolic spaces [7], which leverages their natural ability to learn on data with hierarchical structure. A key measure of characterizing how well a hyperbolic space fits a given dataset is the Gromov hyperbolicity, which quantifies the extent to which a metric space $(X, d_X)$ deviates from a tree metric, with tree structures achieving a value of zero. As such, it has often been used as a proxy for graph dataset suitability to hyperbolic space embedding (*e.g.* [8, 40]), which has also been recently analyzed in [23].

Metrics that can be exactly realized as shortest-path distances in a tree play a central role in many domains: hierarchical clustering in machine learning [28], phylogenetic tree construction in bioinformatics [14], and modeling real-world networks [1, 34], to cite a few. When data is not exactly

39th Conference on Neural Information Processing Systems (NeurIPS 2025).

tree-structured, a natural problem arises: how to approximate an arbitrary metric as faithfully as possible with a tree metric?

Formally, a metric $d_X$ on a finite set $X$ is called a *tree metric* if there exists a tree metric $(T, d_T)$ and an embedding $\Phi : X \to T$ such that

$$d_X(x, y) = d_T(\Phi(x), \Phi(y)) \quad \text{for all } x, y \in X. \tag{1.1}$$

That is, pairwise distances in $X$ can be exactly represented as shortest-path distances in a tree. When such a representation is not possible, one seeks a tree metric that approximates $d_X$ as closely as possible under some distortion measure.

This gives rise to the following problem:

**Problem 1.1** (Minimum Distortion Tree Metric Approximation). Given a finite metric space $(X, d_X)$, our goal is to construct an embedding $\Phi : X \to T$ into a metric tree $(T, d_T)$ such that the tree metric $d_T$ closely approximates the original metric $d_X$. We aim to solve

$$\underset{(T, d_T) \in \mathcal{T}}{\operatorname{argmin}} \| d_X - d_T \|_\infty := \underset{(T, d_T) \in \mathcal{T}}{\operatorname{argmin}} \max_{x, y \in X} |d_X(x, y) - d_T(\phi(x), \phi(y))|,$$

where $\mathcal{T}$ denotes the set of finite tree metrics.

**Tree metrics embedding methods.** In response, several methods have been developed to embed data into tree-like metrics with minimal distortion, e.g. [9, 36, 39]. Among them, the Neighbor Joining algorithm (NJ) [36] has long been a standard algorithm for constructing tree metrics from distance data, which proceeds by iteratively joining neighboring nodes. An alternative, the TREEREP algorithm [39], offers improved scalability. Despite their algorithmic efficiency and empirical success, both methods lack theoretical guarantees regarding the distortion incurred during the embedding process. This limitation affects their reliability in applications where preserving metric fidelity is critical. In contrast, based on a theoretical characterization of the distortion, the GROMOV algorithm [21], described in more details later in the paper, constructs a tree metric by iteratively selecting pairs of points. It serves as a key component of our approach. Restricted to unweighted graph, the LAYERINGTREE method [9] first chooses a root node, then join iteratively neighbors with a bounded additive distortion. The more recent HCCROOTEDTREEFIT algorithm [45] starts by centering the distance matrix around a chosen root, then fits an ultrametric distance to the centered distance, minimizing an average $\ell_1$-distortion. However, its accuracy is generally lower than that of the methods presented above and remains sensitive to the choice of root. Those various approaches are illustrated in Figure 1, along with our new method described subsequently. Additional illustrations on toy graphs are also provided in Appendix A with details on how the figure is generated. Related works in hyperbolic embeddings (e.g., [30, 37]) also aim to capture hierarchical structure, albeit from a different modeling perspective, generally based on contrastive losses that preserve relationships. In contrast, we operate directly on pairwise distances and focus on learning a surrogate metric that can be explicitly embedded into a tree with worst-case distortion guarantees. This metric-centric view differs from position/relation-based hyperbolic embeddings, but we see them as highly complementary.

**Contributions.** In this paper, we propose DELTAZERO, a new method for solving the minimum distortion tree metric approximation problem. It formulates the problem as a constrained optimization task, where the objective is to minimize the distortion between the input metric and a surrogate metric space $(X', d_{X'})$, while enforcing that the resulting space is close to a true tree metric, as quantified by a low Gromov hyperbolicity. This surrogate is eventually embedded into an actual tree $(T, d_T)$. One additional key feature of DELTAZERO is its differentiable formulation: it leverages smooth approximations of both the Gromov hyperbolicity and the distortion to enable gradient-based optimization. Computing the exact Gromov hyperbolicity for a dataset of even moderate size is computationally demanding, both in terms of time and memory consumption. The best known theoretical algorithm operates in $O(n^{3.69})$ time [17], which is impractical for large graphs. Although faster empirical approximations have been proposed, such as [10], scalability remains a bottleneck. To address this, we propose an alternative strategy based on batch-wise sampling. Rather than analyzing the entire dataset at once, DELTAZERO estimates hyperbolicity and distortion on subsets of the distance matrix, which significantly reduces computational overhead while preserving accuracy. We demonstrate that this mini-batching approach enables scalable Gromov hyperbolicity approximation without sacrificing fidelity to the exact value.

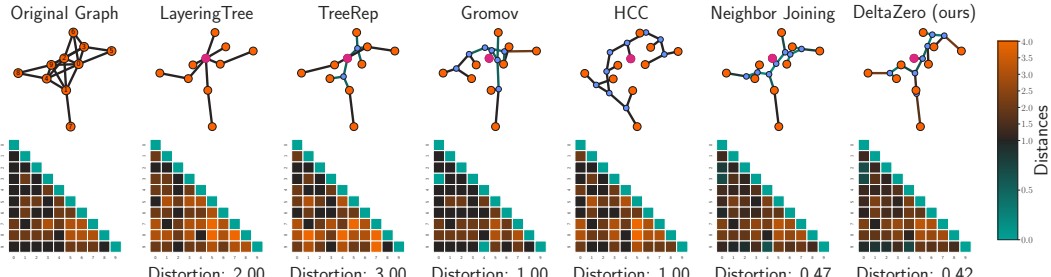

Figure 1: Illustration of the tree metric embedding problem (best viewed with colors). Given an original graph (first column) and the corresponding shortest path distances between nodes (represented on the bottom row as a lower-triangular matrix), we aim at finding a tree and the corresponding tree metric that best approximates the original distances. Competing state-of-the-art methods and our method DELTAZERO results are presented along with the corresponding distortion.

The rest of the paper is organized as follows. Section 2 introduces the notion of $\delta$- and Gromov hyperbolicity and gives intuition on its geometric interpretation. In Section 3, we formulate our optimization problem and show how it improves the worst-case distortion of Gromov. We then provide a differentiable and efficient algorithm to solve the problem. In the experimental Section 4, we first evaluate DELTAZERO in a controlled setting to assess its ability to provide hierarchical clusters. We then measure its ability to generate low distortion tree metric approximations in two contexts, where unweighted and weighted graphs are at stake. Finally, we draw some conclusions and perspectives.

## 2 Background on $\delta$-hyperbolicity

At the heart of our approach lies the concept of $\delta$-hyperbolicity. In this section, we present the foundational principles of this notion and offer intuitive insights into its geometric interpretation. We also discuss relevant computational aspects.

### 2.1 From Gromov product to $\delta$-hyperbolicity

A key concept used to define $\delta$-hyperbolicity is the *Gromov product*, denoted $(x|y)_w$, which intuitively measures the overlap between geodesic paths from a base point $w$ to the points $x$ and $y$. It is defined as follows.

**Definition 2.1** (Gromov Product). Let $(X, d_X)$ be a metric space and let $x, y, w \in X$. The *Gromov product* of $x$ and $y$ with respect to the basepoint $w$ is defined as

$$(x|y)_w = \frac{1}{2} \left( d_X(x, w) + d_X(y, w) - d_X(x, y) \right).$$

With the Gromov product in hand, we now define $\delta$-hyperbolicity.

**Definition 2.2** ($\delta$-hyperbolicity and Gromov hyperbolicity). A metric space $(X, d_X)$ is said to be $\delta$-hyperbolic if there exists $\delta \geq 0$ such that for all $x, y, z, w \in X$, the Gromov product satisfies

$$(x|z)_w \geq \min \left\{ (x|y)_w, (y|z)_w \right\} - \delta.$$

The *Gromov hyperbolicity*, denoted by $\delta_X$, is the smallest value of $\delta$ that satisfies the above property. Consequently, every finite metric space $(X, d_X)$ has a Gromov hyperbolicity equal to

$$\delta_X = \max_{x,y,z,w \in X} \left( \min \left\{ (x|y)_w, (y|z)_w \right\} - (x|z)_w \right). \tag{2.1}$$

The concept of $\delta$-hyperbolicity may initially appear abstract, but it has deep and elegant connections to tree metrics. In fact, a metric $d_X$ is a tree metric if and only if, for every four points $x, y, z, w \in X$, two largest among the following three sums

$$d_X(x, y) + d_X(z, w), \quad d_X(x, z) + d_X(y, w), \quad d_X(x, w) + d_X(y, z)$$

are equal [6].

The $\delta$-hyperbolicity generalizes this condition by allowing a bounded deviation. Specifically, a metric space $(X, d_X)$ is $\delta$-hyperbolic if, for all $x, y, z, w \in X$, the following inequality holds

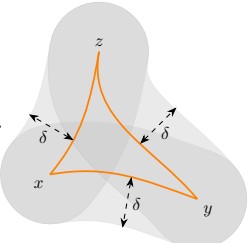

$$d_X(x,y) + d_X(z,w) \leq \max\left\{d_X(x,z) + d_X(y,w), d_X(x,w) + d_X(y,z)\right\} + 2\delta.$$

This is known as the *four-point condition*. It offers an alternative characterization of $\delta$-hyperbolicity (see [19, Ch. 2,§1]), providing a quantitative measure of how far a given metric deviates from being a tree metric, and thereby capturing an intrinsic notion of negative curvature in the space. An alternative and more geometrically intuitive way to understand $\delta$-hyperbolicity is through the concept of $\delta$-*slim geodesic triangles*. In this perspective, a geodesic metric space $(X, d_X)$ is $\delta$-hyperbolic if and only if every geodesic triangle in $X$ is

Figure 2: $\delta$-slim triangle.

$\delta$-slim (up to a multiplicative constant on $\delta$, see [19, Ch.2, §3, Proposition 21]), meaning that each side of the triangle is contained within the $\delta$-neighbourhood of the union of the two other sides (see Figure 2).

This captures the idea that geodesic triangles in hyperbolic spaces are "thin", more closely resembling tripods than the broad, wide-angled triangles characteristic of Euclidean geometry. In this sense, Gromov hyperbolicity quantifies the extent to which the space deviates from tree-like behaviour: the smaller the value of $\delta$, the more the geodesic triangles resemble those in a tree, where all three sides reduce to a union of two overlapping segments. Thus, the $\delta$-slim triangle condition offers a compelling geometric counterpart to the more algebraic formulations via the Gromov product or the four-point condition.

**Computing the Gromov hyperbolicity.**   Computing the Gromov hyperbolicity of a discrete metric space is computationally demanding, as it requires examining all quadruples of points to evaluate the four-point condition. The naive brute-force approach runs in $O(n^4)$ time for a space with $n$ points, making it impractical for large-scale applications. Then, several works have addressed the computational bottlenecks inherent in computing Gromov hyperbolicity. Notably, Fournier et al. [17] show that the computation of hyperbolicity from a fixed base-point can be reduced to a $(\max, \min)$ matrix product. Leveraging the fast algorithm for this class of matrix products, it leads to an overall $O(n^{3.69})$ time complexity.

In graph settings, the notion of *far-apart* vertex pairs plays a key role in accelerating hyperbolicity computation. This concept underpins a structural result [10] stating that certain far-apart pairs suffice to witness the maximum in the definition of Gromov hyperbolicity. Consequently, one can avoid exhaustive examination of all quadruples and focus on a carefully selected subset; this leads to a pruning approach that significantly reduces computational cost [11].

Contrary to these approaches, which aim to accelerate the exact computation of Gromov hyperbolicity, we take a different route: we introduce a smooth, differentiable surrogate of the hyperbolicity function. This relaxation enables gradient-based optimization, and we further propose a batched approximation scheme to make the computation tractable and independent of the size of the graph.

## 2.2   Embedding of a $\delta$-hyperbolic space into a tree

The $\delta$-hyperbolicity can play a significant theoretical role in establishing guarantees for embedding arbitrary metric spaces into trees. Indeed, a result by Gromov [21] shows that any $\delta$-hyperbolic metric space on $n$ points admits a tree-metric approximation with additive distortion $O(\delta \log n)$, and that this embedding can be computed in $O(n^2)$ time.

**Theorem 2.3** ([19, Ch.2, §2, Theorem 12]). *Let $(X, d_X)$ be a finite $\delta$-hyperbolic metric space over $n$ points. For every $w \in X$, there exists a finite metric tree $(T, d_T)$, and a map $\Phi : X \longrightarrow T$ such that*

1. *The distance to the basepoint is preserved: $d_T(\Phi(x), \Phi(w)) = d_X(x, w)$   for all $x \in X$,*

2. $d_X(x,y) - 2\delta \log_2(n-2) \leq d_T(\Phi(x), \Phi(y)) \leq d_X(x,y)$   *for all $x, y \in X$.*

The approximation of distances provided in the previous theorem is particularly notable due to the non-expansiveness of the mapping, meaning that the embedded distances in the tree never exceed the original graph distances. This ensures that the distortion introduced is purely additive and one-sided,

which is a stronger guarantee than general metric embeddings. In [45], Yim et al. provide an argument to illustrate the asymptotic sharpness of the result, using a construction in the Poincaré disk.

For a practical and efficient computational procedure to obtain Gromov's tree embedding of a discrete metric space, one can leverage the *Single Linkage Hierarchical Clustering* (SLHC) algorithm [31]. The SLHC algorithm incrementally merges clusters based on minimal pairwise distances, implicitly constructing an ultrametric tree. While originally developed for clustering tasks, this procedure can be naturally adapted to construct the Gromov's embedding. Further details are provided in Appendix B.

While this result is notable, it provides a worst-case, coarse approximation that does not seek to minimize the actual distortion for a specific input metric. In contrast, our work focuses on directly learning a tree-like metric that is optimally close to the original space, potentially achieving significantly lower distortion in practice, as shown in the experiments, see Section 4.

## 3   DELTAZERO

We present below our approach to the Minimum Distortion Tree Metric Approximation problem. In this work, we consider discrete metric spaces and we denote by $\mathcal{M}_n$ the set of finite metric spaces over $n$ points, and fix $(X, d_X) \in \mathcal{M}_n$. Our algorithm is inspired by Theorem 2.3 which states that $(X, d_X)$ can be isometrically embedded into a tree if and only if its Gromov hyperbolicity is $0$. Building on this result, we introduce the following optimization problem, parametrized by $\mu > 0$,

$$\min_{\substack{(X', d_{X'}) \in \mathcal{M}_n \\ d_{X'} \leq d_X}} \mathcal{L}_X((X', d_{X'}), \mu) := \mu \|d_X - d_{X'}\|_\infty + \delta_{X'}. \tag{3.1}$$

By minimizing $\mathcal{L}_X$, we seek a discrete metric space that is both close to $(X, d_X)$ and exhibits low hyperbolicity. This objective balances two competing goals: proximity to the original metric (via the $\ell_\infty$ distortion term) and tree-likeness (quantified by $\delta_{X'}$). The parameter $\mu$ governs the trade-off between these terms. The constraint $d_{X'} \leq d_X$ accounts for the non-expansiveness of the metric, as in Theorem 2.3 for the Gromov embedding. Importantly, the minimization of $\mathcal{L}_X$ leads to improved worst-case guarantees when followed by the Gromov tree embedding procedure, as described below.

**Theorem 3.1.** *Let $(X^*, d_{X^*})$ be any metric space minimizing $\mathcal{L}_X((X', d_{X'}), \mu)$, and let $(T^*, d_{T^*})$ be a tree metric obtained by applying a Gromov tree embedding denoted by $\Phi$ to $(X^*, d_{X^*})$. Then*

$$d_X(x, y) - C_{X,\mu} \leq d_{T^*}(\Phi(x), \Phi(y)) \leq d_X(x, y) \text{ for all } x, y \in X,$$

*with $C_{X,\mu} = 2\delta_X \log_2(n-2) + (1 - 2\log_2(n-2)\mu) \|d_X - d_{X^*}\|_\infty$. In particular, if $\mu \geq \frac{1}{2\log_2(n-2)}$, the distortion of the final tree embedding is smaller than the worst-case bound achieved by applying the Gromov embedding directly to $X$.*

The proof of all theorems are given in App. C. Theorem 3.1 suggests that improved $\ell_\infty$ distortions can be achieved by minimizing $\mathcal{L}_X$. Our solution, which we describe below, adopts this principle.

### 3.1   Smooth Relaxation and Batched Gromov Hyperbolicity

We propose a differentiable loss function that allows us to optimize the metric so that it closely aligns with the original metric while maintaining a small Gromov hyperbolicity. The primary challenge lies in the non-differentiability of $\delta_X$, as it involves $\max$ and $\min$ operators, rendering it unsuitable for gradient-based optimization. To address this, we introduce a smoothing of Gromov hyperbolicity based on the log-sum-exp function defined as $\text{LSE}_\lambda(\mathbf{x}) = \frac{1}{\lambda} \log\left(\sum_i e^{\lambda x_i}\right)$ for a vector $\mathbf{x}$ and $\lambda > 0$. As done in many contexts, $\text{LSE}_{\pm\lambda}$ can be used as a differentiable surrogate for $\max$ and $\min$ [12, 18, 25], recovering both as $\lambda \to \infty$. We define a smooth version of the $\delta$-hyperbolicity as

$$\delta_X^{(\lambda)} := \text{LSE}_\lambda\left(\{\text{LSE}_{-\lambda}((x|y)_w, (y|z)_w) - (x|z)_w\}_{x,y,z,w \in X}\right)$$
$$= \frac{1}{\lambda} \log \sum_{(x,y,z,w) \in X} \frac{e^{-\lambda(x|z)_w}}{e^{-\lambda(x|y)_w} + e^{-\lambda(y|z)_w}}. \tag{3.2}$$

The rationale behind this formulation is to replace the $\min$ in (2.1) with a soft-minimum $\text{LSE}_{-\lambda}$ and the $\max$ with a soft-maximum $\text{LSE}_\lambda$, which boils down to computing the last quantity. Interestingly, we have the following bounds between the Gromov hyperbolicity and its smoothed counterpart $\delta_X^{(\lambda)}$.

**Proposition 3.2.** *Let $(X, d_X)$ be a finite metric space over $n$ points, for $\lambda > 0$ we have*

$$\delta_X - \frac{\log(2)}{\lambda} \leq \delta_X^{(\lambda)} \leq \delta_X + \frac{4\log(n)}{\lambda}.$$

While the smoothed version $\delta_X^{(\lambda)}$ alleviates the non-differentiability of the classical Gromov hyperbolicity, it still requires evaluating all quadruples of nodes in $X$, resulting in a computational complexity of $O(n^4)$. This remains prohibitive for large number of points. To address this, we introduce a batched approximation of the Gromov hyperbolicity. We sample $K$ independent subsets (without replacement within a subset and no constraint across subsets) $X_m^1, \ldots, X_m^K \subset X$, each containing $m$ points, and compute the local smooth hyperbolicity $\delta_{X_m^i}^{(\lambda)}$ within each batch. Since $\delta_X$ is defined as the maximum over all quadruples in $X$, and each $\delta_{X_m^i}^{(\lambda)}$ approximates the maximum over quadruples in a smaller subset $X_m^i \subset X$, we aggregate these local estimates using another log-sum-exp to obtain a differentiable approximation of the global maximum

$$\delta_{X,K,m}^{(\lambda)} = \mathrm{LSE}_\lambda \left( \delta_{X_m^1}^{(\lambda)}, \cdots, \delta_{X_m^K}^{(\lambda)} \right). \tag{3.3}$$

This two-level smoothing approach allows us to preserve differentiability while reducing the computational complexity from $O(n^4)$ to $O(K \cdot m^4)$, making the estimation of Gromov hyperbolicity tractable.

It is important to note that this sampling strategy does not, in general, guarantee closeness to the true Gromov hyperbolicity. For example, one may construct a metric space composed of a large tree with four additional points connected far from the main structure, although these few points may determine the global hyperbolicity, they are unlikely to appear in small random subsets. Thus, rare yet impactful configurations may be missed, leading to underestimation. Nonetheless, we find that in practice, the batched approximation performs well across a variety of graph instances (see Appendix D.3). In particular, we can show that for some random graph models, our estimator closely tracks the true Gromov hyperbolicity, as formalized below.

We say that a random metric space $(X, d_X) \in \mathcal{M}_n$ is *uniform* if the random distances $d_X(x_i, x_j)$ are i.i.d. with some law $\ell$, called the *law of distances* (see precise Definition C.2).

**Theorem 3.3.** *Let $\ell$ be a probability measure on $\mathbb{R}_+$. Let $(X, d_X)$ be a uniform metric space with with law of distances given by $\ell$. In the regime $K\log(m) \sim \log(n)$ and $\lambda \sim \varepsilon^{-1}\log(n)$ we have*

$$\mathbb{P}\left[ \left| \delta_X - \delta_{X,K,m}^{(\lambda)} \right| \leq \varepsilon \right] \geq 1 - \frac{1}{\varepsilon^{2K} n^{1-o(1)}}.$$

This result establishes that, in the asymptotic regime where $n \to \infty$, our batch estimate closely aligns with the true Gromov hyperbolicity, provided that the number of batches times the logarithm of their size, as well as the smoothing parameter, depend logarithmically on $n$. The assumptions about metric spaces imply independent distances, which can be restrictive. As shown in Appendix C.3, a random graph model satisfying these assumptions is one where shortest-path distances are randomly drawn from a distribution over $\{r, \ldots, 2r\}$, or within a certain regime of Erdős–Rényi graphs. In practice, however, some of our datasets exhibit log-normal distance distributions (see Appendix C.3), suggesting that these assumptions are not overly restrictive. Future work will aim to extend these results to encompass a broader class of random metrics.

## 3.2 Final Optimization Objective

To formalize our algorithm, we start by noting that the set of discrete metrics over $n$ points can be identified with the following subset of $\mathbb{R}^{n \times n}$

$$\mathcal{D}_n := \left\{ \mathbf{D} = (D_{ij}) \in \mathbb{R}_+^{n \times n} : \mathbf{D} = \mathbf{D}^\top, D_{ij} \leq D_{ik} + D_{kj}, D_{ii} = 0 \;\; \forall i, j, k \in \{1, \cdots, n\} \right\}.$$

Each $\mathbf{D} \in \mathcal{D}_n$ encodes the pairwise distances of a metric space over $n$ points, in particular, any metric $d_X$ on a set $X = \{x_1, \ldots, x_n\}$ of size $n$ is fully described by its matrix representation $\mathbf{D}_X = (d_X(x_i, x_j))_{i,j} \in \mathcal{D}_n$. Note that $\mathcal{D}_n$ forms a closed, convex, polyhedral cone generated by the half-spaces imposed by the triangle inequalities.

---

**Algorithm 1** DELTAZERO

---

**Require:** A finite metric space $(X, d_X)$, a designated root $w \in X$, learning rate $\epsilon > 0$, number of batches $K$, batch size $m$, log-sum-exp scale $\lambda > 0$, regularization parameter $\mu > 0$, number of steps $T$

1: Initialize $\mathbf{D}_0 = \mathbf{D}_X$
2: **for** $t = 0$ to $T - 1$ **do**
3:      Randomly pick $K$ batches of $m$ points and compute $\delta_{\mathbf{D}_t, K, m}^{(\lambda)}$
4:      Compute gradient: $\mathbf{G}_t = \nabla L_X(\mathbf{D}_t)$
5:      Optimization step: $\tilde{\mathbf{D}}_{t+1} = \text{ADAM STEP}(\mathbf{D}_t, \mathbf{G}_t, \epsilon)$
6:      Projection step: $\mathbf{D}_{t+1} = \text{FLOYD-WARSHALL}(\tilde{\mathbf{D}}_{t+1})$
7: **end for**
8: **return** GROMOV$(\mathbf{D}_t, w)$

---

In practice, to find a tree-like approximation of $(X, d_X)$, we search for a matrix $\mathbf{D} \in \mathcal{D}_n$ that is close to $\mathbf{D}_X$ while exhibiting low Gromov hyperbolicity. Since Gromov hyperbolicity depends only on the metric, we write $\delta_{\mathbf{D}}$ to emphasize this dependence. Our final optimization objective is

$$\min_{\mathbf{D} \in \mathcal{D}_n} L_X(\mathbf{D}) := \mu \|\mathbf{D}_X - \mathbf{D}\|_2^2 + \delta_{\mathbf{D}, K, m}^{(\lambda)}, \tag{3.4}$$

where $\mathbf{A} \leq \mathbf{B}$ means $\forall (i, j), A_{ij} \leq B_{ij}$. The first term preserves fidelity to the input metric, acting as a smooth counterpart to the $\ell_\infty$ norm. Our choice of the squared $\ell_2$ norm stems from its smoothness and differentiability, which make it well-suited to gradient-based optimization. Moreover, among the common $\ell_p$ norms, $\ell_2$ provides a natural balance: it is closer to $\ell_\infty$ than $\ell_1$, as captured by the classical inequality in finite-dimensional spaces,

$$\|\mathbf{x}\|_\infty \leq \|\mathbf{x}\|_2 \leq \sqrt{n} \|\mathbf{x}\|_\infty \quad \text{for all } \mathbf{x} \in \mathbb{R}^n. \tag{3.5}$$

The second term promotes low-stretch, hierarchical structure using our smooth log–sum–exp approximation of Gromov hyperbolicity. The constraint $\mathbf{D} \in \mathcal{D}_n$ account for the metric requirement. Note that we do not require anymore the non-expansiveness of the metric. While the objective (3.4) includes a convex fidelity term, the hyperbolicity penalty introduces non-convexity. This comes from the structure of the Gromov hyperbolicity functional.

**Proposition 3.4.** *The Gromov hyperbolicity $\delta_{\mathbf{D}}$ seen as a functional over $\mathcal{D}_n$ is piecewise affine but not convex.*

Thus, we expect our smooth surrogate $\delta_{\mathbf{D}, K, m}^{(\lambda)}$ to also inherits this non-convexity, making the overall objective in (3.4) non-convex.

To tackle this optimization problem, we draw inspiration from Projected Gradient Descent (PGD). Specifically, we alternate between taking a plain gradient step of $L_X$ using ADAM [24] (only over the upper-triangular entries of $\mathbf{D}$, due to symmetry and zero diagonal), and projecting the result back onto $\mathcal{D}_n$. Each projection step solves

$$\Pi_{\mathcal{D}_n}(\mathbf{D}) = \underset{\mathbf{D}' \in \mathcal{D}_n}{\operatorname{argmin}} \|\mathbf{D}' - \mathbf{D}\|_2^2.$$

This projection corresponds to a *metric nearness problem*, which we tackle using the FLOYD–WARSHALL algorithm [16]. This algorithm computes all-pairs shortest paths in a graph with edge weights given by $\mathbf{D}$, and as shown in [3, Lemma 4.1], this is equivalent to finding the closest valid metric matrix no greater than $\mathbf{D}$ in each entry (see Section C.5 for a detailed discussion). The projection runs in $O(n^3)$ time and can be interpreted as iteratively "repairing" triangle inequality violations to reach the nearest point in the metric cone with entrywise smaller values.

At the end of the PGD iterations, we obtain a refined metric $\mathbf{D}^*$. To complete the tree approximation, we apply the Gromov tree embedding to $\mathbf{D}^*$, producing a tree metric. The full procedure, which we refer to as DELTAZERO, is summarized in Algorithm 1.

**Computational complexity.** Each gradient step incurs a computational cost of $O(Km^4 + n^3)$, where the $O(n^3)$ term arises from the use of the FLOYD–WARSHALL algorithm. Consequently, the

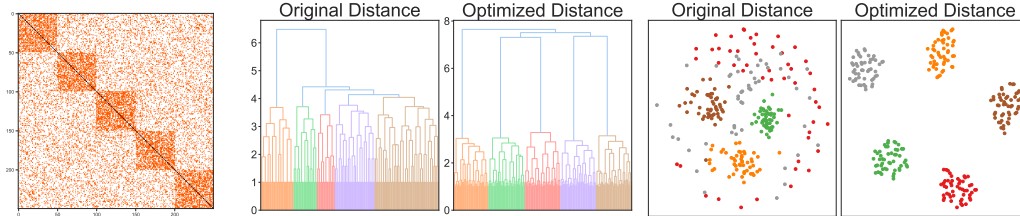

(a) All pairs Shortest-Paths distance matrix

(b) Dendrograms from original and optimized distance matrices

(c) t-SNE plots from original and optimized distance matrices

Figure 3: Illustration of the impact of optimizing the distance matrix on a simple Stochastic Block Model with 5 communities (best viewed in colors).

overall complexity of the optimization procedure over $T$ iterations is $O\big(T(Km^4 + n^3)\big)$. In practice, the computation is typically dominated by the FLOYD–WARSHALL component, as we operate under the condition $Km^4 \ll n^3$. We emphasize that the optimization is carried out in the space of distance matrices, so that the cubic cost of the projection is not disproportionate relative to the size of the object being optimized (with $n^2$ entries). Several methods have been proposed for metric nearness (e.g., [4, 26, 20]), often relying on iterative schemes. Although these approaches may enjoy a lower theoretical complexity, they tend to be slower in practice. This is because FLOYD–WARSHALL, despite its $O(n^3)$ complexity, admits efficient GPU parallelization, leading to substantial speed-ups compared to iterative alternatives.

## 4   Experiments

We provide in this Section experiments conducted on synthetic dataset to assess the quality of the optimization method, and on real datasets to measure the distortion obtained by DELTAZERO.

**Synthetic Stochastic Block Models dataset.** We consider the setting of Hierarchical Clustering (HC) which iteratively merges clusters based on pairwise dissimilarities. The resulting tree-like structure is known as a dendrogram and encodes a hierarchy of nested groupings. Our first objective is to qualitatively and quantitatively analyse the impact of the optimization problem of Eq. (3.4) on HC within the context of a simple stochastic block model (SBM) graph with a varying number of communities.

In such a probabilistic model of graph structure, we set the intra- and inter-communities connection probabilities to $p_{in} = 0.6$ and $p_{out} = 0.2$. Sizes of communities are fixed to 50. This yields a shortest path distance matrix depicted in Figure 3a for a SBM with 5 communities. We then follow the distance optimization procedure described in Section 3.2 to obtain a new distance matrix (parameters are $\mu = 1$ and $\lambda = 100$). This yields two different dendrograms, shown in Figure 3b, both computed using Ward's method [44] as implemented by the `linkage` function of Scipy [43]. The color threshold is manually adjusted to magnify 5 clusters. At this point, we can see that the dendrogram corresponding to the optimized distance exhibit a better, more balanced, hierarchical structure with a clear separation between the 5 clusters. This point is also illustrated in Figure 3c where t-SNE [41] is computed from the distance matrices with a similar perplexity (set to 30). One can observe a stronger clustering effect from the optimized distance.

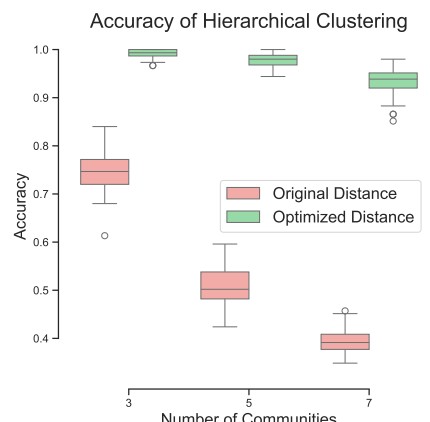

Figure 4: Performances of Hierarchical clustering with varying number of communities

We then compute accuracy scores by comparing the true labels with the optimal permutation of predicted labels, using the `fcluster` function of Scipy. This evaluation is conducted under varying number of communities (3, 5 and 7), introducing increasing levels of difficulty. For each setting, 30 repetitions are performed. The results are summarized as boxplots in Figure 4. As one can observe, our optimization strategy increases drastically the performances in terms of accuracy on this simple example. It suggests that our optimization produces a metric which effectively homogenizes intra- and inter- cluster distances.

**Distortions on real datasets.** We evaluate the ability of our method, DELTAZERO, to embed finite metric spaces into tree metrics with low distortion. Our benchmark includes both unweighted graph datasets and general metric datasets to showcase the versatility of our approach. We consider five standard unweighted graphs: C-ELEGAN, CS PHD from [35], CORA, AIRPORT from [38] and WIKI [15]. For graphs with multiple connected components, we extract the largest connected component. We then compute the shortest-path distance matrix, which serves as input to all tree metric fitting algorithms.

To evaluate DELTAZERO beyond graph-induced metrics, we also include two datasets: ZEISEL [46] and IBD [32]. These datasets are not naturally represented as graphs but instead as high-dimensional feature matrices. We construct a pairwise dissimilarity matrix using cosine distance, a standard choice in bioinformatics. Despite its name, cosine distance is not a true metric, as it fails to satisfy the triangle inequality, but it still provides a meaningful basis from which to search for the closest tree metric. Note that LAYERINGTREE [9], which requires an unweighted graph as input, is not applicable in this setting. We compare DELTAZERO against the following tree fitting methods: TREEREP (TR) [39], NEIGHBORJOIN (NJ) [36], HCCROOTEDTREEFIT (HCC) [45], LAYERINGTREE (LT) [9], and the classical GROMOV (see Algorithm 2). Among these, HCC, LT, and GROMOV are pivot-based methods: for a given distance matrix, they require selecting a root node. We report the average and standard deviation of distortion over 100 runs using the same randomly sampled roots across methods. TREEREP does not require a root but incorporates stochastic elements; we therefore report mean and standard deviation over 100 independent runs. For C-ELEGAN, CS PHD, CORA, and AIRPORT, we report the values of [45]. In contrast, NEIGHBORJOIN is deterministic that neither requires a root nor involves stochasticity, and is thus evaluated once per dataset.

For DELTAZERO, we perform grid search over the following hyperparameters: learning rate $\epsilon \in \{0.1, 0.01, 0.001\}$, distance regularization coefficient $\mu \in \{0.1, 0.01, 1.0\}$, and $\delta$-scaling parameter $\lambda \in \{0.01, 0.1, 1.0, 10.0\}$. We fix the number of training epochs to $T = 1000$, batch size $m = 32$, and vary the number of batches $K \in \{100, 500, 1000, 3000, 5000\}$. For each setting, we select the best configuration which leads to the minimal distortion. Note that Theorem 3.1 offers theoretical guidance on worst-case distortion guarantees, suggesting $\mu \geq 1/\log_2(n-2)$. In practice, however, we treat $\mu$ as a tunable hyperparameter, reflecting the fact that we optimize smooth surrogates rather than enforce exact hard constraints.

To ensure stability, we apply early stopping with a patience of 50 epochs and retain the model with the best training loss. In practice, our method converges in a reasonable number of iterations; detailed early stopping statistics are reported in Appendix D.1.

Final results are reported in Table 1. In addition to worst-case distortion defined in equation (1.1), we evaluate the average embedding quality using the $\ell_1$ distance between the original and tree-fitted distance matrices. We also report the execution time of each method to assess their computational efficiency. Results are presented in Appendix D.2. Note that, as highlighted by equation (3.5), in finite-dimensional spaces, minimizing the $\ell_2$ norm offers no direct control over the $\ell_\infty$ norm, which can lead, during optimization, to lower average distortion but potentially higher worst-case distortion. We also provide an analysis of its robustness and stability with respect to hyperparameter in Appendix D.3. All implementations details are given in Appendix E.

Table 1 demonstrates that DELTAZERO consistently achieves the lowest $\ell_\infty$ distortion across all datasets, both for unweighted graphs and general metric spaces. Notably, the largest relative improvements are observed on C-ELEGAN and ZEISEL, with reductions in distortion of 43.8% and 44.1% respectively, compared to the second-best methods. On CORA the improvement is more modest (2.3%), yet DELTAZERO still outperforms all baselines. This suggests that while our approach is robust across datasets, gains vary depending on the geometry of the input metric space. Overall, the

Table 1: $\ell_\infty$ error (lower is better). Best result in bold. The last row reports the relative improvement (%) of DELTAZERO over the second-best method (underlined) for each dataset.

| Datasets | Unweighted graphs | | | | | Non-graph metrics | |
|---|---|---|---|---|---|---|---|
| | C-ELEGAN | CS PHD | CORA | AIRPORT | WIKI | ZEISEL | IBD |
| $n$ | 452 | 1025 | 2485 | 3158 | 2357 | 3005 | 396 |
| Diameter | 7 | 28 | 19 | 12 | 9 | 0.87 | 0.99 |
| $\delta_X$ | 1.5 | 6.5 | 4 | 1 | 2.5 | 0.19 | 0.41 |
| NJ | $\underline{2.97}$ | 16.81 | 13.42 | 4.18 | 6.32 | 0.51 | $\underline{0.90}$ |
| TR | $5.90 \pm 0.72$ | $21.01 \pm 3.34$ | $16.86 \pm 2.11$ | $10.00 \pm 1.02$ | $9.97 \pm 0.93$ | $0.66 \pm 0.10$ | $1.60 \pm 0.22$ |
| HCC | $4.31 \pm 0.46$ | $23.35 \pm 2.07$ | $12.28 \pm 0.96$ | $7.71 \pm 0.72$ | $7.20 \pm 0.60$ | $0.53 \pm 0.07$ | $1.25 \pm 0.11$ |
| LayeringTree | $5.07 \pm 0.25$ | $25.48 \pm 0.60$ | $\underline{7.76} \pm 0.54$ | $\underline{2.97} \pm 0.26$ | $\underline{4.08} \pm 0.27$ | – | – |
| Gromov | $3.33 \pm 0.45$ | $\underline{13.28} \pm 0.61$ | $9.34 \pm 0.53$ | $4.08 \pm 0.27$ | $5.54 \pm 0.49$ | $\underline{0.43} \pm 0.02$ | $1.01 \pm 0.04$ |
| DELTAZERO | $\mathbf{1.87} \pm 0.08$ | $\mathbf{10.31} \pm 0.62$ | $\mathbf{7.59} \pm 0.38$ | $\mathbf{2.79} \pm 0.15$ | $\mathbf{3.56} \pm 0.20$ | $\mathbf{0.24} \pm 0.00$ | $\mathbf{0.70} \pm 0.03$ |
| Improvement (%) | 43.8% | 22.3% | 2.3% | 6.0% | 12.7% | 44.1 % | 22.2% |

results validate the effectiveness of our optimization-based method in producing low-distortion tree metric embeddings.

## 5 Conclusion and Discussion

While our primary focus has been on distance approximation, we believe our method has strong potential for standard graph-representation tasks. By producing a surrogate distance metric that captures global and hierarchical structure in a differentiable, geometry-aware manner, our framework provides a rich structural signal that can be exploited across multiple settings. For example, one could embed our optimized metric into hyperbolic spaces to obtain high-quality node embeddings, or feed it directly to downstream neural models.

In node classification, the tree-like structures induced by our method may uncover latent hierarchies or community structure, an intuition validated by our experiments on stochastic block models, where DELTAZERO helps accurately recovering the underlying partitions. Beyond synthetic settings, this makes our approach especially well-suited for real-world discovery tasks such as biological taxonomies, document hierarchies, or social networks. Extending our method to such domains would be natural continuation of the present work.

Our framework can also be integrated as a pre-processing step or regularizer in graph neural networks, where it provides additional structural bias. This is particularly relevant for addressing the over-squashing phenomenon: Gromov hyperbolicity is tied to negative sectional curvature, while over-squashing has been linked to negative Ollivier–Ricci curvature [29]. By bridging these notions of curvature, our method could offer a principled avenue to investigate, and potentially mitigate such effects, opening the door to curvature-aware re-wiring procedures and architectural design.

## Acknowledgements

This project was supported by the ANR project OTTOPIA ANR-20-CHIA-0030. The authors declare no competing financial interests.

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

## A  Complementary qualitative and visual results

We provide in Figure 5 complementary qualitative results obtained on simple unweighted graphs, and details about the generation procedure. The original graph is an unweighted graph, with constant unit edge weights, generated following different graph models: Figures 5a and 5b correspond to random connected Erdős–Rényi graph with 10 nodes, Figure 5c is a stochastic block model graph with 2 communities and 40 nodes, generated with an intra-community probability of $0.5$ and an inter-community probability of $0.05$, and finally Figure 5d is a $(4, 4)$-grid graph, with a total of 16 nodes. The original nodes are depicted in orange in the figures, whereas additional nodes, when provided by the method, are depicted in blue. The root node used for generating the corresponding tree is depicted in pink, and is set to be the same for all the methods. In every visualizations, the original nodes positions are kept fixed, and the added nodes positions are computed using the SPRINGLAYOUT from the NetworkX package [22].

In terms of distortion, our method gives the best results in 3 out of 4 cases, and ranks second in the grid-like graph case, where LAYERINGTREE performs best.

## B  Gromov's embedding

The tree metric $d_T$, obtained by the embedding $\Phi$ given by Theorem 2, is characterized via Gromov products with respect to a basepoint $w$, which we denote by

$$(x|y)'_w := \frac{1}{2}\left(d_T(\Phi(x), \Phi(w)) + d_T(\Phi(y), \Phi(w)) - d_T(\Phi(x), \Phi(y))\right).$$

This yields the identity

$$d_T(\Phi(x), \Phi(y)) = (x|x)_w + (y|y)_w - 2(x|y)'_w.$$

Gromov's argument [21, Page 156] further shows that the Gromov product $(x|y)'_w$ can be expressed as

$$(x|y)'_w = \max_{\bar{y} \in P_{x,y}} \left\{ \min_k (y_k|y_{k+1})_w \right\},$$

where $P_{x,y}$ denotes the set of finite sequences $\bar{y} = (y_1, \ldots, y_\ell)$ in $X$ with $\ell \geq 2$, $x = y_1$, and $y = y_\ell$.

It is important to note that in Gromov's construction, the resulting tree metric space $(T, d_T)$ may include intermediate points that are not present in the original set $X$, but are introduced in the tree to satisfy the metric constraints and to preserve approximate distances (see Figure 6). These points serve as internal branching nodes that allow better approximation of the original distances under the tree metric.

To approximate an optimal ultrametric tree embedding of a finite metric space $(X, d)$, one can leverage the *single linkage hierarchical clustering* (SLHC) algorithm, which is known to yield a 2-approximation to the optimal ultrametric tree embedding [31]. The SLHC algorithm returns a new metric $u_X$ on $X$, defined for all $x, y \in X$ by:

$$u_X(x, y) = \min_{\bar{y} \in P_{x,y}} \left\{ \max_k d(y_k, y_{k+1}) \right\},$$

where $P_{x,y}$ denotes the set of finite sequences $\bar{y} = (y_1, \ldots, y_\ell)$ such that $y_1 = x$, $y_\ell = y$, and $y_i \in X$ for all $i$. To relate this to a tree embedding via Gromov products, fix a base point $w \in X$, and define

$$m := \max_{x \in X} d(x, w).$$

Then the function $d_G : X \times X \to \mathbb{R}$ given by

$$d_G(x, y) := \begin{cases} m - (x|y)_w & \text{if } x \neq y, \\ 0 & \text{if otherwise} \end{cases}$$

defines a valid metric on $X$. Applying SLHC to the metric space $(X, d_G)$ yields an ultrametric space $(X, u_X)$. Finally, we define the tree metric as

$$d_T(x, y) := m - d_G(x, y).$$

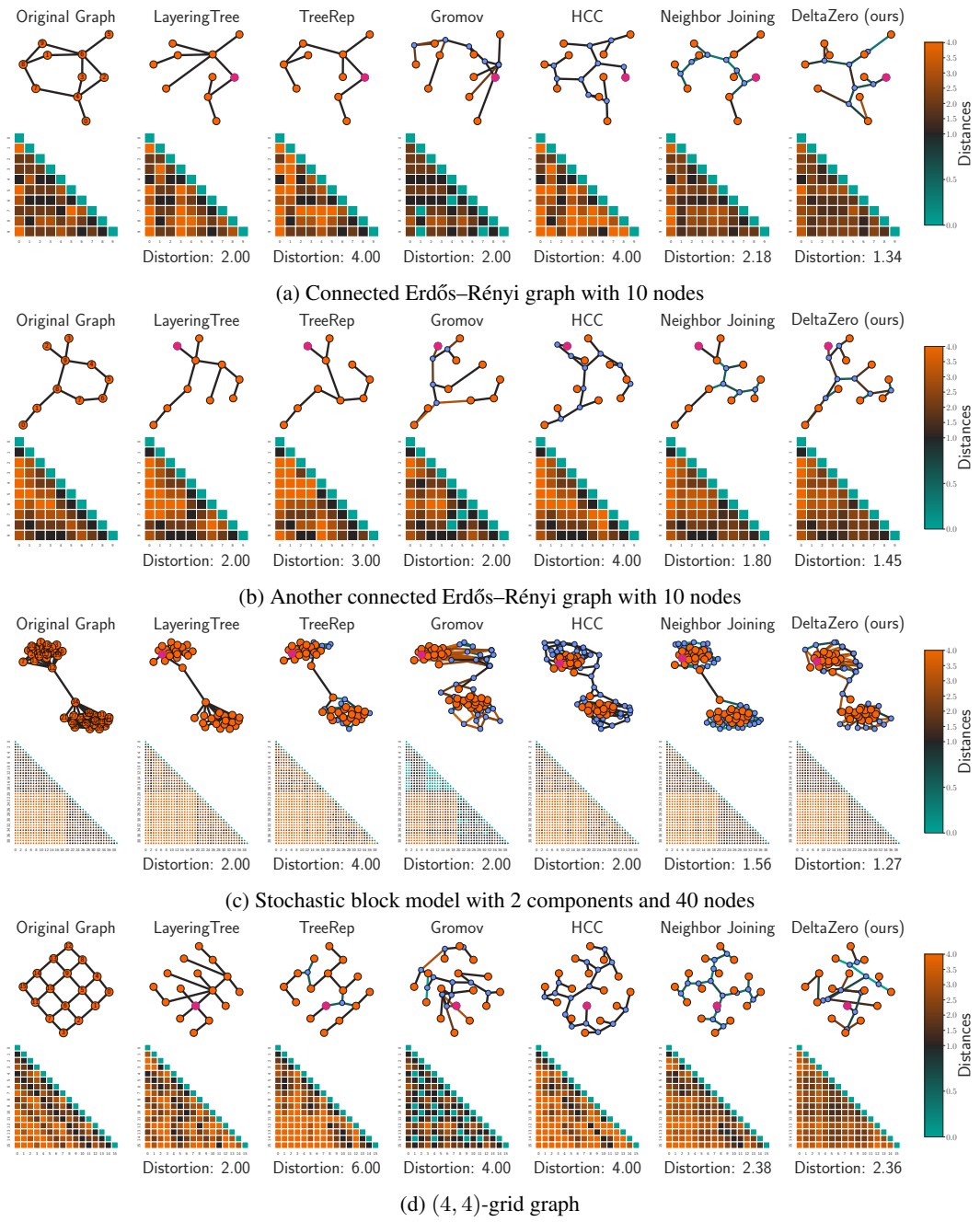

(a) Connected Erdős–Rényi graph with 10 nodes

(b) Another connected Erdős–Rényi graph with 10 nodes

(c) Stochastic block model with 2 components and 40 nodes

(d) $(4, 4)$-grid graph

Figure 5: Visual illustration of the different methods on different graphs.

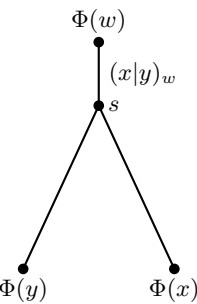

Figure 6: Gromov tree embedding of the points in Euclidean space: $x = (1, 0)$, $y = (-1, 0)$, and $w = (0, 3)$. A intermediate node $s$ is introduced at a position such that it lies at distance $(x|y)_w$ from $w$.

One can easily check that this procedure gives the same tree metric than the embedding given by Gromov.

---

**Algorithm 2** GROMOV

---

**Require:** Finite metric space $(X, d)$, base point $w \in X$
**Ensure:** Tree metric $d_T$ on $X$
  1: Compute Gromov products: $(x|y)_w \leftarrow \frac{1}{2}(d(x, w) + d(y, w) - d(x, y))$
  2: Set $m \leftarrow \max_{x \in X} d(x, w)$
  3: Define $d_G(x, y) \leftarrow m - (x|y)_w$ if $x \neq y$, 0 otherwise
  4: Apply SLHC to $(X, d_G)$ to obtain an ultrametric $u_X$
  5: Define tree metric: $d_T(x, y) \leftarrow m - u_X(x, y)$
  6: **return** $d_T$

---

Once the tree metric has been obtained through this procedure, one may then invoke the method described in [6] to reconstruct a tree spanning the set $X$. This reconstruction yields both the intermediate nodes and the leaves, the latter corresponding to all elements of $X$ except the designated root.

## C  Proofs of Theoretical Results

### C.1  Proof of Theorem 3.1

For a fixed $\mu$, let $(X^*, d_{X^*})$ be a minimizer of (3.1). We have, for any $(X, d_X)$,

$$\mu\|d_X - d_{X^*}\|_\infty + \delta_{X^*} \leq \delta_X.$$

Applying the Gromov tree embedding $\Phi$ to $(X^*, d_{X^*})$ and denoting the resulting metric tree by $(T^*, d_{T^*})$ we get the following bound using Theorem 2.3

$$d_{X^*}(x, y) - 2(\delta_X - \mu\|d_X - d_{X^*}\|_\infty)\log_2(n-2) \leq d_{T^*}(\Phi(x), \Phi(y)) \leq d_{X^*}(x, y) \text{ for all } x, y \in X.$$

Combining this with the triangle inequality yields

$$
\begin{aligned}
\|d_X - d_{T^*}\|_\infty &\leq \|d_X - d_{X^*}\|_\infty + \|d_{X^*} - d_{T^*}\|_\infty \\
&\leq \|d_X - d_{X^*}\|_\infty + 2(\delta_X - \mu\|d_X - d_{X^*}\|_\infty)\log_2(n-2) \\
&\leq 2\delta_X\log_2(n-2) + (1 - 2\log_2(n-2)\mu)\|d_X - d_{X^*}\|_\infty.
\end{aligned}
$$

Thus, we have

$$d_X(x, y) - (2\delta_X\log_2(n-2) + (1 - 2\log_2(n-2)\mu)\|d_X - d_{X^*}\|_\infty) \leq d_{T^*}(\Phi(x), \Phi(y)).$$

For the second inequality, we have $\forall x, y,\ d_T(\Phi(x), \Phi(y)) \leq d_{X^*}(x, y) \leq d_X(x, y)$. The first inequality is true by Gromov theorem and the second due to the constraints in the optimization problem (3.1).

As long as $\mu \geq \frac{1}{2 \log_2(n-2)}$, the worst-case distortion achieved by applying the Gromov embedding directly to $X$ is improved.

## C.2   Proof of Proposition 3.2

To control the approximation error introduced by the smooth maximum operator, we first state a standard inequality satisfied by the $\text{LSE}_\lambda$ function.

**Proposition C.1.** *For $\lambda > 0$, the $\text{LSE}_\lambda$ function satisfies the following inequalities*

$$\max \mathbf{x} \leq \text{LSE}_\lambda(\mathbf{x}) \leq \max \mathbf{x} + \frac{\log(n)}{\lambda},$$

$$\min \mathbf{x} - \frac{\log(n)}{\lambda} \leq \text{LSE}_{-\lambda}(\mathbf{x}) \leq \min \mathbf{x}.$$

*Proof.* Let $m = \max_i x_i$. Then we have

$$e^{\lambda m} \leq \sum_{i=1}^n e^{\lambda x_i} \leq n e^{\lambda m}.$$

Taking the logarithm and dividing by $\lambda > 0$, we obtain

$$m \leq \text{LSE}_\lambda(\mathbf{x}) \leq m + \frac{\log(n)}{\lambda}. \qquad \square$$

We are now ready to use this bound to control the smooth approximation of Gromov's four-point condition and prove proposition 3.2.

In $X$, there are $n^4$ ordered 4-tuples of points with replacement. By proposition C.1, for any fixed $x, y, z, w$, we have

$$\min\{(x|y)_w, (y|z)_w\} - (x|z)_w - \frac{\log(2)}{\lambda} \leq \text{LSE}_{-\lambda}((x|y)_w, (y|z)_w) - (x|z)_w$$

and

$$\max_{x,y,z,w} \{\text{LSE}_{-\lambda}((x|y)_w, (y|z)_w) - (x|z)_w\} \leq \delta_X^{(\lambda)} \leq \delta_X + \frac{4 \log(n)}{\lambda}.$$

Combining those inequalities, we obtain the desired result.

## C.3   Proof of Theorem 3.3

For every $n \in \mathbb{N}$, let $X = \{x_1, \ldots, x_n\}$ be a fixed $n$-point set.

We consider a *random metric space* on $n$ points. This means we fix a probability measure $\nu$ on $\mathcal{M}_n$, the set of all metric spaces over $n$ points.

Since a metric space on $n$ points is completely determined by the distances between each pair of points, we can think of $\nu$ as a probability law on the set of all possible distance matrices that satisfy the metric conditions (non-negativity, symmetry, triangle inequality, and zero diagonal).

So, when we say that a random variable $X$ has law $\nu$, we mean that we are choosing the distances between the $n$ points at random, according to $\nu$. In this way, the law $\nu$ is a *law on the distances* between the points.

**Definition C.2.** We say that a random metric space $(X, d)$ (with law $\nu$) is *uniform* if the random variables $d(x_i, x_j)$ for $x_i, x_j \in X$ are independent and identically distributed, following a common law $\ell$. We say that $\ell$ is the *law of distances* of the uniform random metric space.

In this class of random metric spaces, we have the following asymptotics for $\delta_X^{(\lambda)}$.

**Proposition C.3.** *Let $\ell$ be a probability measure on $\mathbb{R}_+$ (with compact support). Let $\nu_n$ be a probability measure on $\mathcal{M}_n$ such that the associated random metric space is uniform, with law of*

*distances given by $\ell$. Let $(X, d)$ be a random metric following the law $\nu_n$. Then, when $n$ goes to infinity and $\lambda = O(\log n)$, we have*

$$\mathbb{P}\left( \left| \delta_X^{(\lambda)} - \frac{1}{\lambda}(\log \mu + 4 \log n) \right| \geq \varepsilon \right) \leq \frac{1}{\varepsilon^2 n^{1-o(1)}},$$

*where $\mu$ depends on the law $\ell$ and $\lambda$, but not on $n$ and $\varepsilon$.*

*Proof of Proposition C.3.* We use the standard convention that the letter $C$ denotes a universal constant, whose precise value may change from line to line.

Recall from (3.2) that $\delta_X^{(\lambda)}$ can be written explicitly as

$$\delta_X^{(\lambda)} = \frac{1}{\lambda} \log \sum_{(x,y,z,w) \in X^4} \frac{e^{-\lambda(x|z)_w}}{e^{-\lambda(x|y)_w} + e^{-\lambda(y|z)_w}},$$

where the Gromov product are taken with respect to the metric $d$. We denote by $f(x, y, z, w)$ the term

$$\frac{e^{-\lambda(x|z)_w}}{e^{-\lambda(x|y)_w} + e^{-\lambda(y|z)_w}}.$$

We start by a preliminary estimate on $f$. We introduce some notations. We note

$$\delta_{x,y,z,w} = \min((x|y)_w, (y|z)_w) - (x|z)_w.$$

We say that a quadruple $q = (x, y, z, w) \in X^4$ is *generic* if the elements $x, y, z, w$ are pairwise distinct. Observe that if $q_1$ and $q_2$ are both generic, then $f(q_1)$ and $f(q_2)$ are identically distributed. In particular,

$$\mathbb{E}[f(q_1)] = \mathbb{E}[f(q_2)],$$

and we denote this common expectation by $\mu$.

**Lemma C.4.** *For every $p \in \mathbb{N}$, we have the following asymptotics when $\lambda$ goes to infinity*

$$\mathbb{E}\left[ f(q)^p \right]^{\frac{1}{p}} = e^{\lambda \max \delta_q} e^{o(\lambda)},$$

*and*

$$\mathbb{E}\left[ f(q)^p \right]^{\frac{1}{p}} \leq \mu e^{o(\lambda)}.$$

*Proof.* Recall that

$$f(x, y, z, w) = \frac{e^{-\lambda(x|z)_w}}{e^{-\lambda(x|y)_w} + e^{-\lambda(y|z)_w}} = \exp\left( \lambda(\mathrm{LSE}_{-\lambda}((x|y)_w, (y|z)_w) - (x|z)_w) \right),$$

hence by Proposition C.1:

$$\exp\left( \lambda \left( \delta_{x,y,z,w} - \frac{\log 2}{\lambda} \right) \right) \leq f(x, y, z, w) \leq \exp\left( \lambda \delta_{x,y,z,w} \right).$$

We conclude that

$$C_1 e^{\lambda \delta_q} \leq f(q) \leq C_2 e^{\lambda \delta_q},$$

for universal constants $C_1, C_2$.

We recall the classical limit

$$\lim_{\lambda \to \infty} \left( \mathbb{E}\left[ X^\lambda \right] \right)^{\frac{1}{\lambda}} = \max X,$$

which holds for a positive, bounded random variable, which we can rewrite

$$\mathbb{E}\left[ X^\lambda \right] = (\max X)^\lambda e^{o(\lambda)}.$$

Applying that to $X = e^{p\delta_q}$ we obtain

$$\left( \mathbb{E}\left[ e^{\lambda p \delta_q} \right] \right)^{\frac{1}{p}} = e^{\lambda \max \delta_q} e^{o(\lambda)},$$

and thus

$$\left(\mathbb{E}\left[f(q)^p\right]\right)^{\frac{1}{p}} = e^{\lambda \max \delta_q} e^{o(\lambda)}.$$

Observe that the law of the random variable $\delta_q$, for $q = (x, y, z, w)$, depends only of the configuration of which $x, y, z, w$ are equal between them. In particular, there is a finite number of possibilities for $\max \delta_q$ (which depend only on the law $\ell$). It is tedious but easy to check that $\max \delta_q$ is maximum when $q$ is generic. Recalling that $\mu$ is a notation for $\mathbb{E}\left[f(q)\right]$ when $q$ is generic, we conclude that:

$$\left(\mathbb{E}\left[\bar{f}(q)^p\right]\right)^{\frac{1}{p}} \leq \mu e^{o(\lambda)}. \qquad \square$$

We now study the sum

$$S_n = \sum_{(x,y,z,w) \in X^4} f(x, y, z, w).$$

If the terms of the sum were independent random variables, we could directly apply standard limit theorems. However, this is not the case: each term $f(x, y, z, w)$ depends only on the pairwise distances between the four points $x, y, z, w$. In particular, two terms $f(x, y, z, w)$ and $f(x', y', z', w')$ are not independent whenever the sets of their arguments intersect, i.e.,

$$\{x, y, z, w\} \cap \{x', y', z', w'\} \neq \emptyset.$$

To formalize this, for two 4-tuples $q_1 = (x, y, z, w)$ and $q_2 = (x', y', z', w')$ in $X_n^4$, we write $q_1 \perp q_2$ to denote that they are disjoint

$$\{x, y, z, w\} \cap \{x', y', z', w'\} = \emptyset.$$

Despite the dependencies, the situation remains favorable. The number of dependent pairs,

$$\{(q_1, q_2) \in (X^4)^2 : q_1 \not\perp q_2\},$$

is bounded by $Cn^7$, while the total number of pairs in $(X^4)^2$ is $n^8$. Thus, most of the terms are approximately independent, which allows for concentration techniques and asymptotic analysis.

We adapt the classical proof of the weak law of large numbers to this weakly dependent setting using Chebyshev's inequality. To that end, we compute the variance of the sum

$$S_n = \sum_{(x,y,z,w) \in X_n^4} f(x, y, z, w).$$

It is convenient to consider the centered version

$$\bar{S}_n = \sum_{(x,y,z,w) \in X_n^4} \bar{f}(x, y, z, w), \quad \text{where} \quad \bar{f}(x, y, z, w) = f(x, y, z, w) - \mathbb{E}[f(x, y, z, w)],$$

so that $\mathbb{E}[\bar{S}_n] = 0$. The variance is then

$$\mathbb{E}[\bar{S}_n^2] = \sum_{(q_1, q_2) \in (X^4)^2} \mathbb{E}[\bar{f}(q_1)\bar{f}(q_2)]$$

$$= \sum_{q_1 \perp q_2} \mathbb{E}[\bar{f}(q_1)\bar{f}(q_2)] + \sum_{q_1 \not\perp q_2} \mathbb{E}[\bar{f}(q_1)\bar{f}(q_2)].$$

Since $q_1 \perp q_2$ implies independence and $\mathbb{E}[\bar{f}(q_1)] = 0$, the first sum vanishes. Thus, we are left with

$$\mathbb{E}[\bar{S}_n^2] = \sum_{q_1 \not\perp q_2} \mathbb{E}[\bar{f}(q_1)\bar{f}(q_2)].$$

By Cauchy-Schwarz inequality we have

$$\mathbb{E}\left[\bar{f}(q_1)\bar{f}(q_2)\right] \leq \left(\mathbb{E}\left[\bar{f}(q_1)^2\right]\mathbb{E}\left[\bar{f}(q_2)^2\right]\right)^{\frac{1}{2}},$$

and by the triangle inequality

$$\left(\mathbb{E}\left[\bar{f}(q)^2\right]\right)^{\frac{1}{2}} = \mathbb{E}\left[(f(q) - \mathbb{E}[f(q)])^2\right]^{\frac{1}{2}} \leq \mathbb{E}\left[f(q)^2\right]^{\frac{1}{2}} + \mathbb{E}\left[f(q)\right].$$

Applying the Lemma, both terms on the right hand side are bounded by $\mu e^{o(\lambda)}$, hence

$$(\mathbb{E}\left[\bar{f}(q)^2\right])^{\frac{1}{2}} \leq \mu e^{o(\lambda)}.$$

Using that the number of pairs with $q_1 \not\perp q_2$ is bounded by $Cn^7$ we finally obtain the variance bound

$$\mathbb{E}[\bar{S}_n^2] \leq \mu^2 e^{o(\lambda)} n^7.$$

We recall Chebyshev's classical inequality:

$$\mathbb{P}(|X - \mathbb{E}[X]| \geq \varepsilon) \leq \frac{\mathbb{E}[(X - \mathbb{E}[X])^2]}{\varepsilon^2}.$$

Applying this inequality to $X = \bar{S}_n$ and $\varepsilon' = n^4 \varepsilon$ we get

$$\mathbb{P}\left(|\bar{S}_n| \geq n^4 \varepsilon\right) \leq \frac{\mathbb{E}[\bar{S}_n^2]}{\varepsilon^2 n^8},$$

and using our previous estimate

$$\mathbb{P}\left(\left|\frac{1}{n^4}\bar{S}_n\right| \geq \varepsilon\right) \leq \frac{\mu^2 e^{o(\lambda)}}{\varepsilon^2 n} = \frac{\mu^2}{\varepsilon^2 n^{1-o(1)}},$$

which we can rewrite as

$$\mathbb{P}\left(\left|\frac{1}{n^4}S_n - \frac{1}{n^4}\mathbb{E}[S_n]\right| \geq \varepsilon\right) \leq \frac{\mu^2}{\varepsilon^2 n^{1-o(1)}}.$$

We now compute the term $\frac{1}{n^4}\mathbb{E}[S_n] = \frac{1}{n^4}\sum_{q \in X^4}\mathbb{E}[f(q)]$. We can write:

$$\frac{1}{n^4}\mathbb{E}[S_n] = \frac{1}{n^4}\sum_{q \in X^4}\mathbb{E}[f(q)]$$

$$= \frac{1}{n^4}\sum_{\substack{q \in X^4 \\ \text{generic}}}\mathbb{E}[f(q)] + \frac{1}{n^4}\sum_{\substack{q \in X^4 \\ \text{non-generic}}}\mathbb{E}[f(q)]$$

Moreover, the number of non-generic quadruples (i.e., $q \in X^4$ where at least two of the entries coincide) is at most $Cn^3$ for some constant $C > 0$. By the Lemma, $\mathbb{E}[f(q)]$ is bounded by $\mu e^{o(\lambda)}$. Combining these observations, we obtain

$$\frac{1}{n^4}\mathbb{E}[S_n] = \left(1 - O\left(\frac{1}{n}\right)\right)\mu + \frac{\mu e^{o(\lambda)}}{n} = \mu\left(1 + \frac{1}{n^{1-o(1)}}\right)$$

which implies that $\frac{1}{n^4}\mathbb{E}[S_n] \sim \mu$ as $n \to \infty$.

We can now look at $\delta_X^{(\lambda)} = \frac{1}{\lambda}\log S_n$. We recall the elementary inequality

$$|\log a - \log b| \leq \frac{|a - b|}{b}.$$

If $|S_n - \mathbb{E}[S_n]| \leq n^4 \varepsilon$, then

$$\left|\frac{S_n - \mathbb{E}[S_n]}{\mathbb{E}[S_n]}\right| \leq \frac{n^4 \varepsilon}{\mathbb{E}[S_n]}.$$

Since $\mathbb{E}[S_n] \sim n^4 \mu$, we have for large $n$,

$$\frac{1}{\mathbb{E}[S_n]} \leq \frac{2}{n^4 \mu}.$$

Hence, for $n$ large enough,

$$|\log S_n - \log \mathbb{E}[S_n]| \leq \frac{|S_n - \mathbb{E}[S_n]|}{\mathbb{E}[S_n]} \leq \frac{2\varepsilon}{\mu}.$$

Therefore, the log-difference is controlled by the relative deviation scaled by $\mu^{-1}$.

We conclude that when $n$ is large enough,

$$\mathbb{P}\left(\left|\log S_n - \log \mathbb{E}[S_n]\right| \geq \frac{2\varepsilon}{\mu}\right) \leq \mathbb{P}\left(\left|\frac{1}{n^4}S_n - \frac{1}{n^4}\mathbb{E}[S_n]\right| \geq \varepsilon\right) \leq \frac{\mu^2}{\varepsilon^2 n^{1-o(1)}},$$

which implies

$$\mathbb{P}\left(\left|\frac{1}{\lambda}\log S_n - \frac{1}{\lambda}\log \mathbb{E}[S_n]\right| \geq \varepsilon\right) \leq \frac{1}{\lambda^2\varepsilon^2 n^{1-o(1)}} \leq \frac{1}{\varepsilon^2 n^{1-o(1)}}.$$

Using that $\frac{1}{n^4}\mathbb{E}[S_n] \sim \mu$, when $n$ is large enough we then have:

$$\mathbb{P}\left(\left|\frac{1}{\lambda}\log S_n - \frac{1}{\lambda}(\log \mu + 4\log n)\right| \geq \varepsilon\right) \leq \frac{1}{\varepsilon^2 n^{1-o(1)}}. \qquad \square$$

We can finally state the result that justify the batching method used to compute the delta.

Given a random metric space on $n$ points, distributed according to a law $\nu$, we can define a random metric space on $m \leq n$ points by restriction. Let $(X, d)$ be a random metric space drawn from $\nu$ and let $X_m \subset X$ be a subset of size $m$, chosen uniformly at random. We equip $X_m$ with the metric $d_{|X_m}$ induced by $d$. We can iterate this process $k$-times, choosing the subsets of size $m$ disjoint from each other, and we obtain a sequence of random metric space $X_m^1, \ldots, X_m^K$, each one uniform with law of distances $\ell$.

**Theorem C.5.** *Let $\ell$ be a probability measure on $\mathbb{R}_+$. Let $\nu$ be a probability measure on $\mathcal{M}_n$ such that the associated random metric space is uniform, with law of distances given by $\ell$. In the regime $m^K \sim n$ and $\lambda \sim \varepsilon^{-1}\log n$ we have*

$$\mathbb{P}\left[\exists i, \left|\delta_X^{(\lambda)} - \delta_{X_m^i}^{(\lambda)}\right| \leq \varepsilon\right] \geq 1 - \frac{1}{\varepsilon^{2K}n^{1-o(1)}}$$

*Proof.* For a uniform random metric space $(X, d)$ and a sequence of random subpaces $(X_m^1, \ldots, X_m^K)$ as constructed above, each $(X_m^i, d_{|X_m^i})$ is still a uniform random metric space with the same law of distances. Because $m^K \sim n$ and $\lambda = O(\log n)$ we have $\lambda = O(\log m)$. We apply the Proposition C.3 to get

$$\mathbb{P}\left(\left|\delta_X^{(\lambda)} - \frac{1}{\lambda}(\log \mu + 4\log n)\right| \geq \varepsilon\right) \leq \frac{1}{\varepsilon^2 n^{1-o(1)}},$$

$$\mathbb{P}\left(\left|\delta_{X_m^i}^{(\lambda)} - \frac{1}{\lambda}(\log \mu + 4\log m)\right| \geq \varepsilon\right) \leq \frac{1}{\varepsilon^2 n^{1-o(1)}}.$$

The $(Y_m^i)$ are independent random variables hence

$$\mathbb{P}\left(\forall i \leq K, \left|\delta_{X_m^i}^{(\lambda)} - \frac{1}{\lambda}(\log \mu + 4\log m)\right| \leq \varepsilon\right) \leq \frac{1}{\varepsilon^{2K}m^{K-o(1)}},$$

which means that

$$\mathbb{P}\left(\exists i \leq K, \left|\delta_{X_m^i}^{(\lambda)} - \frac{1}{\lambda}(\log \mu + 4\log m)\right| \leq \varepsilon\right) \geq 1 - \frac{1}{\varepsilon^{2K}m^{K-o(1)}} = 1 - \frac{1}{\varepsilon^{2K}n^{1-o(1)}},$$

When

$$\left|\delta_{X_m^i}^{(\lambda)} - \frac{1}{\lambda}(\log \mu + 4\log m)\right| \leq \varepsilon,$$

and

$$\left|\delta_X^{(\lambda)} - \frac{1}{\lambda}(\log \mu + 4\log n)\right| \leq \varepsilon.$$

Thus,

$$\left|\delta_X^{(\lambda)} - \delta_{X_m^i}^{(\lambda)}\right| \leq 2\varepsilon + \frac{1}{\lambda}\log\frac{n}{m} \leq 3\varepsilon.$$

We can conclude that

$$\mathbb{P}\left[\exists i, \left|\delta_X^{(\lambda)} - \delta_{X_m^i}^{(\lambda)}\right| \leq \varepsilon\right] \geq 1 - \mathbb{P}\left[\forall i, \left|\delta_X^{(\lambda)} - \delta_{X_m^i}^{(\lambda)}\right| \geq \varepsilon\right]$$

$$- \mathbb{P}\left(\left|\delta_X^{(\lambda)} - \frac{1}{\lambda}(\log\mu + 4\log n)\right| \geq \varepsilon\right)$$

$$\geq 1 - \frac{1}{\varepsilon^{2K} n^{1-o(1)}}.$$

$\square$

**Corollary C.6.** *Let $\ell$ be a probability measure on $\mathbb{R}_+$. Let $\nu$ be a probability measure on $\mathcal{M}_n$, the set of metric spaces over $n$ points, such that the associated random metric space is uniform (see definition C.2), with law of distances given by $\ell$. In the regime $m^K \sim n$ and $\lambda \sim \varepsilon^{-1}\log n$ we have*

$$\mathbb{P}\left[\left|\delta_X - \delta_{X,K,m}^{(\lambda)}\right| \leq \varepsilon\right] \geq 1 - \frac{1}{\varepsilon^{2K} n^{1-o(1)}}.$$

*Proof.* This follows directly from triangular inequality, definition of $\delta_{X,K,m}^{(\lambda)}$ (see equation (3.3)) and Proposition 3.2. $\square$

**Models of uniform random metric spaces.** We now describe several models of random metric spaces that are (aproximatively) uniform.

In [27] the authors study a model of discrete random metric spaces. For an integer $r \geq 2$, consider the finite subset $\mathcal{D}_n^r \subset \mathcal{D}_n$ of metric spaces for which the distance function takes values in $\{0, 1, \ldots, 2r\}$. Let $\nu_n^r$ be the uniform probability on this finite set. Let $C_n^r \subset \mathcal{D}_n^r$ the subset of metric spaces whose distance function takes values in the interval $\{r, \ldots, 2r\}$. Observe that any collection of numbers in $\{r, \ldots, 2r\}$ satisfy the triangle inequality, hence $C_n^r = \{r, \ldots, 2r\}^{\binom{n}{2}}$, and the restriction of the measure $\nu_n^r$ to $C_n^r$ is simply a product measure. In other words, the random metric space conditioned to be in $C_n^r$ is uniform. The authors prove that, when $n$ is large, the measure $\nu_n^r$ is exponentialy concentrated on $C_n^r$.

**Theorem C.7.** *There exists $\beta > 0$ such that for $n$ large enough we have*

$$\nu_n^r(C_n^r) \geq 1 - e^{-\beta n}.$$

In other words, the uniform random metric space with integer valued distances and diameter bounded by $2r$ is (approximatively) uniform.

Let $G(n, p)$ denote the Erdős–Rényi random graph: a graph on $n$ vertices where each edge between a pair of vertices is included independently with probability $p \in [0, 1]$.

In the regime

$$p^2 n - 2\log n \to \infty,$$
$$n^2(1 - p) \to \infty,$$

the graph $G(n, p)$ asymptotically almost surely has *diameter 2*, that is, any pair of vertices is either directly connected by an edge or shares a common neighbor, by [2]. According to [13, §4], conditionnaly to the diameter being 2, the distance matrix of this graph is a function of the incidence matrix; in particular, the associated random metric space is uniform.

We also evaluate the distribution of distances on real datasets as shown in Figure 7. We can see that approximately these distributions are log-normal, suggesting that they could satisfy the hypothesis of our theorem.

## C.4 Proof of Proposition 3.4

Following Bridson and Haefliger [5], the four points condition given in (2.1), can be seen in the following way: Let $\mathbf{D} \in \mathcal{D}_n$, for $i, j, k, l \in \{1, \ldots, n\}$, let $l_1, l_2$ and $l_3$ be defined by

$$\begin{cases} l_1 = D_{ij} + D_{kl} \\ l_2 = D_{ik} + D_{jl} \\ l_3 = D_{il} + D_{jk} \end{cases}$$

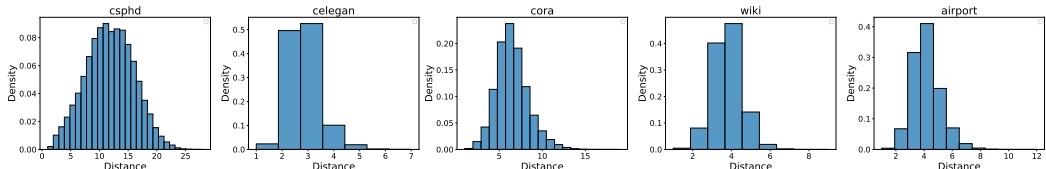

Figure 7: Distribution of the distances of the unweighted datasets of our benchmark.

and let $M_1$ and $M_2$ be the two largest values among $l_1, l_2$, and $l_3$. We define

$$\delta(i, j, k, l) = M_1 - M_2$$

and the Gromov hyperbolicity $\delta_{\mathbf{D}}$ of the metric space is the maximum of $\delta(i, j, k, l)$ over all possible 4-tuples $(i, j, k, l)$ divided by 2. That is,

$$\delta_{\mathbf{D}} = \frac{1}{2} \max_{i,j,k,l \in \{1,\dots,n\}} \delta(i, j, k, l)$$

The function $\delta_{\mathbf{D}}$, viewed as a function over $\mathcal{D}_n$ is a maximum of piecewise affine functions, and is therefore itself piecewise affine.

We can interpret $\delta_{\mathbf{D}}$ geometrically as follows. Each 4-tuple $(i, j, k, l)$ determines three quantities $l_1, l_2$, and $l_3$. The differences between these quantities partition the plane into regions corresponding to which of $l_1, l_2, l_3$ is largest, second largest, and smallest (see Figure 8).

To demonstrate the non-convexity of the set in question, it suffices to exhibit a single counterexample. We do so by constructing two metric spaces, each defined over four points, and represented via their respective distance matrices. These examples are deliberately simple to allow for full transparency of the argument and ease of verification.

While we restrict ourselves here to the case $n = 4$, the same construction strategy can be generalized without difficulty to any number of points $n \geq 4$. The underlying mechanism responsible for the failure of convexity is not peculiar to this specific dimension.

Let us consider

$$\mathbf{D}_1 = \begin{bmatrix} 0 & 1.1 & 1 & 1.2 \\ & 0 & 1 & 1 \\ & & 0 & 1 \\ & & & 0 \end{bmatrix} \text{ and } \mathbf{D}_2 = \begin{bmatrix} 0 & 1 & 1 & 1.2 \\ & 0 & 1 & 1.1 \\ & & 0 & 1 \\ & & & 0 \end{bmatrix}.$$

Computing the associated triples $(l_1, l_2, l_3)$ for each space, we find

$$L_1 = (2.1, 2.2, 2) \quad \text{for} \quad \mathbf{D}_1, \qquad L_2 = (2, 2.2, 2.1) \quad \text{for} \quad \mathbf{D}_2.$$

Now consider the interpolation between $\mathbf{D}_1$ and $\mathbf{D}_2$ defined by $t \in [0, 1] \mapsto t\mathbf{D}_1 + (1 - t)\mathbf{D}_2$ with associated triple $(l_1(t), l_2(t), l_3(t))$.

Since the values of $l_2$ are identical for $\mathbf{D}_1$ and $\mathbf{D}_2$, the interpolation affects only $l_1$ and $l_3$. Thus, along this interpolation $\delta$ behave like

$$t \mapsto -\max(l_1(t), l_3(t)),$$

which is concave and negates that $\delta_{\mathbf{D}}$ is a convex function over $\mathcal{D}_4$.

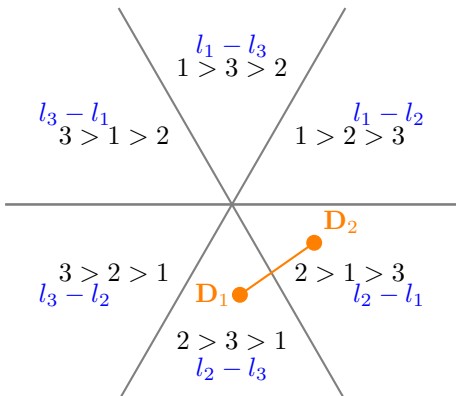

Figure 8: In the diagram, the plane is divided into six sectors, each labeled according to the ordering of $l_1, l_2, l_3$ (e.g., $1 > 2 > 3$ means $l_1$ is largest, then $l_2$, then $l_3$). The labels inside each sector indicate the expression that represents $\delta(i, j, k, l)$ in that region.

## C.5 Floyd-Warshall as a projection

This section mainly rewrites results described in [3]. Let $\mathbf{W} \in \mathbb{R}_+^{n \times n}$ be the matrix of non-negative edge weights of a (possibly directed) graph with $n$ nodes. Let $\mathrm{SP}(\mathbf{W})$ denote the matrix of shortest path distances between all node pairs. We use the notation $\mathbf{A} \leq \mathbf{B}$ to mean that $A_{ij} \leq B_{ij}$ for all $(i, j)$, and recall that $\mathcal{D}_n$ denotes the set of $n \times n$ distance matrices (i.e., matrices that satisfy the properties of a metric).

We aim to show that

$$\mathrm{SP}(\mathbf{W}) \in \underset{\mathbf{D} \in \mathcal{D}_n, \, \mathbf{D} \leq \mathbf{W}}{\operatorname{argmin}} \|\mathbf{D} - \mathbf{W}\|_p,$$

for any $\ell_p$ norm with $p > 0$.

Note that $\mathrm{SP}(\mathbf{W}) \leq \mathbf{W}$ since the shortest path between two nodes is always less than or equal to the direct edge weight (if it exists). Hence, $\mathrm{SP}(\mathbf{W})$ is a feasible candidate for the minimization problem.

Suppose the following property holds:

$$\forall \mathbf{D} \in \mathcal{D}_n, \ \mathbf{D} \leq \mathbf{W} \Rightarrow \mathbf{D} \leq \mathrm{SP}(\mathbf{W}).$$

Under this assumption, the optimality of $\mathrm{SP}(\mathbf{W})$ follows immediately. Indeed, for any such $\mathbf{D}$, we have:

$$0 \leq W_{ij} - \mathrm{SP}(\mathbf{W})_{ij} \leq W_{ij} - D_{ij} = |D_{ij} - W_{ij}|,$$

for all $(i, j)$, using the facts that $\mathrm{SP}(\mathbf{W}) \leq \mathbf{W}$, $\mathbf{D} \leq \mathrm{SP}(\mathbf{W})$, and $\mathbf{D} \leq \mathbf{W}$. Therefore,

$$|\mathrm{SP}(\mathbf{W})_{ij} - W_{ij}| \leq |D_{ij} - W_{ij}|,$$

which implies

$$\|\mathrm{SP}(\mathbf{W}) - \mathbf{W}\|_p \leq \|\mathbf{D} - \mathbf{W}\|_p.$$

We now prove the property. Assume for contradiction that there exists $(i, j)$ such that $D_{ij} > \mathrm{SP}(\mathbf{W})_{ij}$. Consider the lex smallest such pair under the total ordering induced by increasing values of $\mathrm{SP}(\mathbf{W})_{ij}$. That is, sort all node pairs so that:

$$\mathrm{SP}(\mathbf{W})_{i_1, j_1} \leq \mathrm{SP}(\mathbf{W})_{i_2, j_2} \leq \cdots \leq \mathrm{SP}(\mathbf{W})_{i_M, j_M},$$

and let $(i, j)$ be the first pair for which $D_{ij} > \mathrm{SP}(\mathbf{W})_{ij}$.

There are two possibilities: (1) $\mathrm{SP}(\mathbf{W})_{ij} = W_{ij}$. Then $D_{ij} > W_{ij}$, which contradicts the assumption $\mathbf{D} \leq \mathbf{W}$. (2) There exists some node $k$ such that

$$\mathrm{SP}(\mathbf{W})_{ij} = \mathrm{SP}(\mathbf{W})_{ik} + \mathrm{SP}(\mathbf{W})_{kj}.$$

Since $\mathbf{D}$ is a metric, it satisfies the triangle inequality:

$$D_{ij} \leq D_{ik} + D_{kj}.$$

Table 2: Epochs until early stopping for each dataset.

| Datasets | Unweighted graphs | | | | | Non-graph metrics | |
|---|---|---|---|---|---|---|---|
| | C-ELEGAN | CS PHD | CORA | AIRPORT | WIKI | ZEISEL | IBD |
| Epochs until Early Stopping | 166 | 402 | 474 | 877 | 1000 | 456 | 122 |

But by minimality of $(i, j)$ in the sorted list, we know:

$$D_{ik} \leq \text{SP}(\mathbf{W})_{ik}, \quad D_{kj} \leq \text{SP}(\mathbf{W})_{kj}.$$

Therefore,

$$D_{ij} \leq D_{ik} + D_{kj} \leq \text{SP}(\mathbf{W})_{ik} + \text{SP}(\mathbf{W})_{kj} = \text{SP}(\mathbf{W})_{ij},$$

which contradicts the assumption that $D_{ij} > \text{SP}(\mathbf{W})_{ij}$.

This contradiction implies that no such $(i, j)$ exists, and thus $\mathbf{D} \leq \text{SP}(\mathbf{W})$, completing the proof.

# D   Complementary experimental results

## D.1   Convergence of the method

In this section, we examine the numerical convergence of our method Table 2 gives the number of epochs for which it has reached the early stopping criterion. These results show that convergence typically occurs well before early stopping (defined as 50 iterations without improvement of the objective) well before the maximum of 1000 epochs, except in the WIKI dataset, suggesting that the optimization process is generally stable and effective. We also show in Figure 9 the evolution of the objective function and its components over training, using the best hyperparameters for each dataset.

## D.2   Average $\ell_1$ error and Runtime Efficiency

In this section, we complement the $\ell_\infty$ distortion results (Table 1) with an analysis of the average embedding quality using $\ell_1$ error. Table 3 reports the mean $\ell_1$ distortion ($\|d - d_T\|_1 / \binom{n}{2}$) between the input distance matrix and the distance matrix induced by the fitted tree metric. This metric captures the average deviation rather than the worst-case distortion, providing a different perspective on the fidelity of the embedding. The experimental setup is identical to that used for Table 1, including the selection of hyperparameters (based on lowest $\ell_\infty$ error) and root sampling strategy.

As shown in Table 3, NEIGHBORJOIN (NJ) achieves the lowest average distortion across all datasets, including both graph-based and non-graph metrics. This is expected, as NJ is a greedy algorithm designed to minimize additive tree reconstruction error from a distance matrix. However, it does not guarantee low worst-case distortion, and its performance in terms of $\ell_\infty$ error is less consistent (see Table 1).

By contrast, our method, DELTAZERO, is not designed to minimize average error directly but rather optimizes a distortion-aware loss function. Despite this, it performs competitively in $\ell_1$ distortion, especially on datasets such as C-ELEGAN and ZEISEL. However, on larger datasets like CORA and AIRPORT, it yields higher average distortion than NJ, suggesting that there may be a trade-off between minimizing $\ell_\infty$ and $\ell_1$ error in practice.

Table 4 reports the average runtime (in seconds) over 10 runs for each method and dataset. For DELTAZERO, we report the runtime of the optimization step only, using the best-performing hyperparameter configuration (as selected in Table 1). For NJ and TREEREP, the reported times exclude the $O(n^3)$ time required to construct a full pairwise distance matrix, as these methods operate directly produce a tree structure.

As expected, DELTAZERO incurs higher computational cost than the baselines, particularly on large datasets such as ZEISEL and CORA, where optimization can take several thousand seconds. This is the price of our optimization method althought as highlighted in Section 3.2, the major bottleneck comes from the usage of FLOYD-WARSHALL algorithm at each step of our proposed gradient descent. Lightweight methods such as GROMOV and HCC run much faster, though they offer less accuracy in both $\ell_\infty$ and $\ell_1$ metrics.

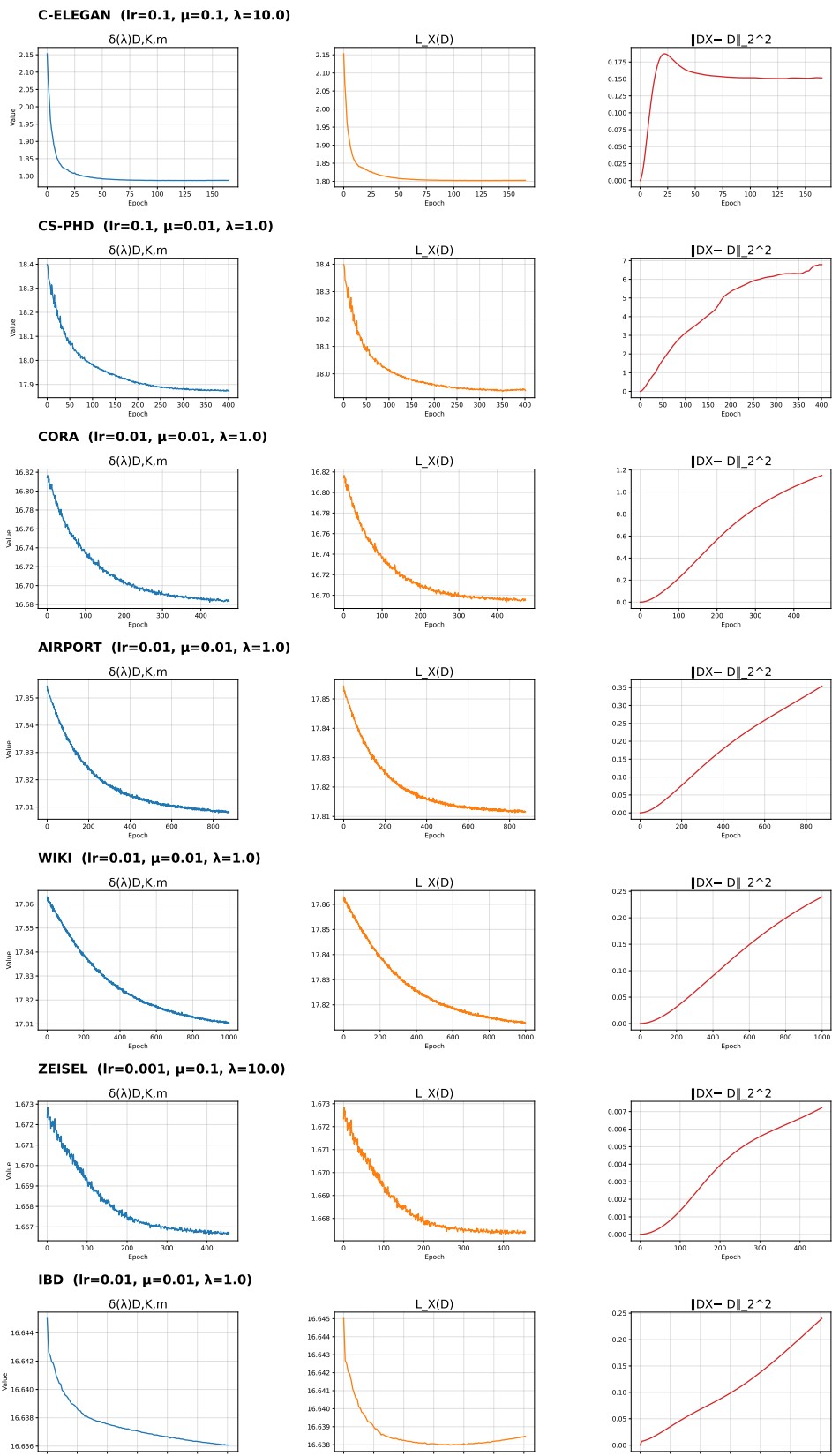

Figure 9: Training metrics over epochs for each dataset.

Table 3: $\ell_1$ average error ($\|d - d_T\|_1 / \binom{n}{2}$). Best result in bold (lower is better).

| Data set | Unweighted graphs | | | | | Non-graph metrics | |
| | C-ELEGAN | CS PHD | CORA | AIRPORT | WIKI | ZEISEL | IBD |
|---|---|---|---|---|---|---|---|
| NJ | **0.30** | **1.35** | **1.06** | **0.49** | **0.51** | **0.04** | 0.09 |
| TR | 0.83 ± 0.16 | 2.55 ± 1.34 | 2.91 ± 0.63 | 1.28 ± 0.21 | 1.85 ± 0.26 | 0.11 ± 0.02 | 0.13 ± 0.01 |
| HCC | 0.88 ± 0.22 | 2.59 ± 1.11 | 2.44 ± 0.43 | 1.10 ± 0.13 | 1.53 ± 0.17 | 0.08 ± 0.02 | 0.27 ± 0.03 |
| LayeringTree | 0.73 ± 0.03 | 4.90 ± 0.07 | 3.68 ± 0.23 | 0.62 ± 0.06 | 0.83 ± 0.04 | – | – |
| Gromov | 1.13 ± 0.01 | 2.80 ± 0.36 | 3.34 ± 0.12 | 1.55 ± 0.08 | 2.12 ± 0.04 | 0.10 ± 0.01 | 0.39 ± 0.02 |
| DELTAZERO | 0.42 ± 0.01 | 2.71 ± 0.08 | 3.20 ± 0.09 | 1.31 ± 0.03 | 1.26 ± 0.00 | 0.08 ± 0.00 | 0.36 ± 0.02 |

Table 4: Average running time in seconds over 10 runs.

| Data set | Unweighted graphs | | | | | Non-graph metrics | |
| | C-ELEGAN | CS PHD | CORA | AIRPORT | WIKI | ZEISEL | IBD |
|---|---|---|---|---|---|---|---|
| NJ | 0.34 | 3.42 | 103.81 | 244.93 | 81.89 | 207.91 | 0.25 |
| TR | 0.78 | 2.64 | 25.64 | 36.37 | 24.43 | 94.02 | 1.23 |
| HCC | 0.67 | 3.54 | 19.54 | 31.15 | 18.28 | 36.21 | 0.52 |
| LayeringTree | 0.86 | 8.81 | 156.54 | 344.43 | 140.11 | – | – |
| Gromov | 0.03 | 0.23 | 1.47 | 2.20 | 24.43 | 2.20 | 0.02 |
| DELTAZERO | 51.00 | 134.28 | 3612.35 | 2268.72 | 819.90 | 5664.40 | 289.07 |

These results highlight a key trade-off: DELTAZERO offers superior worst-case distortion, but at the cost of increased runtime.

**Remark D.1.** Note that we do not report other metrics such as Mean Average Precision (MAP) score [30] (see [42, Equation 33] for the formula), as it does not measure metric fidelity (e.g., uniform rescaling preserves MAP), but only local neighborhood preservation. In our setting, an optimal tree can induce a markedly different neighborhood structure from the original graph (e.g., embedding a grid into a tree). Moreover, MAP is only meaningful when all methods embed into the same target space, which is not the case here. Finally, MAP can only be computed when the metric arises from an underlying graph structure, which is not true for all datasets we consider.

### D.3 Sensitivity analysis

In this section, we provide a sensitivity analysis on the parameters of DELTAZERO, namely the number of batches $K$, the batch size $m$, the temperature parameter $\lambda$ (cf. eq (3.2)) and the distance regularization coefficient $\mu$ on the norm (cf. eq (3.4)).

**Evaluation of the approximation $\delta_{\mathbf{D},K,m}^{(\lambda)}$ of the Gromov hyperbolicity.** We first provide a numerical study of the approximation $\delta_{\mathbf{D},K,m}^{(\lambda)}$ (see eq.(3.3)) of the actual $\delta$-hyperbolicity. We consider the CS-PHD dataset, which has an exact value of $\delta = 6.5$ that can be computed exactly as $n$ is not too large. We evaluate the quality of our approximation with respect to this ground truth.

Figure 10a illustrates the influence of two key hyperparameters: the number of batches $K$ and the batch size $m$, fixing $\lambda = 1000$. Figure 10b illustrates the influence of the regularization parameter $\lambda$ and the batch size $m$, fixing $K = 50$. Both figures also report the true $\delta$. Each point represents the average over 5 independent runs.

We observe that batch size $m$ plays a critical role in the quality of the estimation. When $m$ is very small (e.g., $m = 4$), the estimated values are significantly biased and far from the true $\delta$. As $m$ increases, the estimates become more stable and converge toward the ground truth, even with a moderate number of batches. This pattern is consistent across both panels of the figure. In contrast, the effect of $\lambda$ is less pronounced once it reaches a sufficient scale, beyond which the estimates flatten out near the true $\delta$ value. These results highlight that batch size is the dominant factor influencing the accuracy of our smoothed estimator $\delta_{\mathbf{D},K,m}^{(\lambda)}$. Indeed, computing the exact $\delta$ requires evaluating all possible quadruples of nodes, which amounts to $\binom{n}{4} = O(n^4)$ operations. In our

batched approximation, we instead sample $K$ batches of size $m$, resulting in a total of $K \cdot \binom{m}{4}$ sampled quadruples. Since $\binom{m}{4} = O(m^4)$, the number of evaluated quadruples scales as $O(K \cdot m^4)$, highlighting that $m$ has a higher-order impact than $K$ on the estimator's fidelity. This explains why increasing the batch size leads to more accurate and stable $\delta$ estimates, even with a relatively small number of batches.

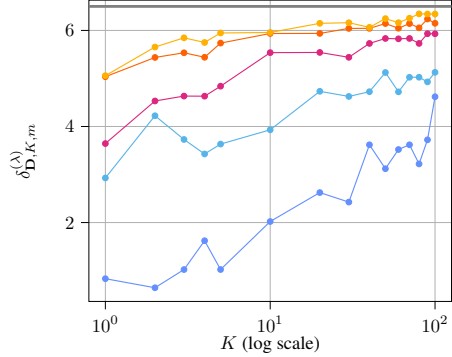
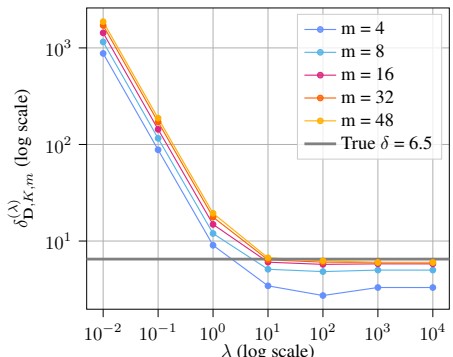

(a) Evolution of the mean $\delta_{\mathbf{D},K,m}^{(\lambda)}$ as a function of the number of batches $K$, with $\lambda = 1000$.

(b) Estimation of $\delta_{\mathbf{D},K,m}^{(\lambda)}$ as a function of $\lambda$, with $K = 50$.

Figure 10: Approximation $\delta_{\mathbf{D},K,m}^{(\lambda)}$ computed on the CS-PHD dataset for different batch sizes $m$.

**Sensitivity to parameters on the distortion.** We now consider the C-ELEGAN dataset and study the impact of the parameter tuning for $\mu$, $\lambda$ and $m$. We vary one hyperparameter at a time, while fixing the others to their optimal values obtained from the grid search. We evaluate the mean distortion over 100 randomly sampled root nodes. This procedure was repeated across 5 independent runs and we plot the mean and standard deviation of the $\ell_\infty$ distortion across these repetitions.

We first study the influence of the regularization coefficient $\mu$ on the $\ell_\infty$ distortion in Figure 11a. One can notice that, for the lowest $\mu$ values, the distortion is high, indicating that the solution is insufficiently driven by the data; for the highest values, the distortion is also high, which suggest that the tree-likeness of the solution is lost. For a good tradeoff between proximity to the original metric and tree-likeness of the solution, an optimum is reached. This further showcases the interest of introducing a term related to the Gromov hyperbolicity into the tree metric approximation problem.

We then study the impact of the $\lambda$ parameter in Figure 11b, for different batch sizes. Results suggest that a good trade-off between large values (that provide an estimate close to the actual value but at the price of sharp optimization landscape) and small ones (that provides a smooth estimate easy to optimize) should be found to reach the best performances.

Finally, in Figure 11c, we study the impact of the batch size $m$, for various $\lambda$ parameters. One can note that, for moderate values of $\lambda$, an optimal performance is reached for a moderate size of batch size, which suggests that $m$ could be set with a relatively low value, enhancing the ability of DELTAZERO to tackle moderated-size datasets.

# E   Implementation details

Our optimization procedure was implemented in Python using `PyTorch` [33] for automatic differentiation and gradient-based optimization. All experiments were run using PyTorch's GPU backend with CUDA acceleration when available. We leveraged the `Adam` optimizer and parallelized computations over batches.

The training objective minimizes a regularized combination of the $\delta$-hyperbolicity loss and the reconstruction error of the distance matrix. The optimization is performed over the upper-triangular weights of a complete undirected graph, which are projected onto shortest-path distances using the Floyd–Warshall algorithm at each iteration. This projection is implemented using a GPU-compatible variant of the algorithm.

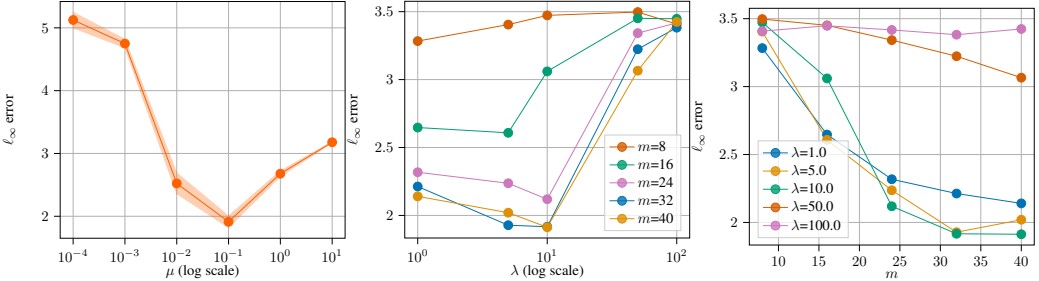

(a) Effect of the distance regulariza-(b) Impact of the log-sum-exp scale (c) Impact of the batch size $m$ on
tion coefficient $\mu$ on $\ell_\infty$ distortion. $\lambda$ on $\ell_\infty$ error across various batch $\ell_\infty$ error for multiple $\lambda$ values
sizes $m$.

Figure 11: Sensitivity analysis of optimization hyperparameters on the C-ELEGAN dataset. In each plot, non-varied hyperparameters are set to their optimal values from a prior grid search. Results are averaged over 5 runs, and distortion values averaged over 100 root samples per run.

To efficiently approximate the Gromov hyperbolicity objective, we randomly sample $K$ batches of quadruples and accumulate gradients across 32 sub-batches in each outer iteration. This simulates a larger batch size while remaining within GPU memory constraints. Gradient accumulation is performed manually by computing the backward pass over each sub-batch before the optimizer step.

Training includes early stopping based on the total loss, with a patience of 50 epochs. We store and return the best-performing weights (in terms of the objective loss (3.4)) encountered during the optimization.

To ensure reproducibility, we provide all code and experiments at `https://github.com/pierrehouedry/DifferentiableHyperbolicity`.

## E.1 Hardware setup

All experiments were conducted using a single NVIDIA TITAN RTX GPU with 24 GB of VRAM. The implementation was written in Python 3.11 and executed on a machine running Ubuntu 22.04. No distributed or multi-GPU training was used.

## E.2 Datasets

For C-ELEGAN and CS PHD, we relied on the pre-computed distance matrices provided at `https://github.com/rsonthel/TreeRep`. The C-ELEGAN dataset captures the neural connectivity of the *Caenorhabditis elegans* roundworm, a widely studied model organism in neuroscience. The CS PHD dataset represents a co-authorship network among computer science PhD holders, reflecting patterns of academic collaboration. For CORA and AIRPORT, we used the graph data from `https://github.com/HazyResearch/hgcn` and computed the shortest-path distance matrices on the largest connected components. CORA is a citation network of machine learning publications, while AIRPORT models air traffic routes between airports. The WIKI dataset, representing a hyperlink graph between Wikipedia pages, was obtained from the `torch_geometric` library [15]. ZEISEL is a single-cell RNA sequencing dataset describing the transcriptomic profiles of mouse cortex and hippocampus cells, commonly used to study cell type organization in neurobiology. It was obtained from `https://github.com/solevillar/scGeneFit-python`. The IBD dataset (Inflammatory Bowel Disease) contains 396 metagenomic samples across 606 expressed microbial species. The dataset is publicly available via BioProject accession number PRJEB1220.

