# OpenReview forum: "Bridging Arbitrary and Tree Metrics via Differentiable Gromov Hyperbolicity"
_NeurIPS.cc/2025/Conference — NeurIPS 2025 poster_

### Official Review · Reviewer_XSQu · 2025-06-30

**Clarity:** 2
**Significance:** 3
**Originality:** 3
**Rating:** 4
**Confidence:** 4

**Summary:**

This paper proposes a tree metric learning method based on Gromov hyperbolicity. The authors begin by formulating the problem as minimum distortion tree metric approximation under the $\ell_\infty$ norm. Then, they introduce a smooth relaxation using the log-sum-exp function, along with a batched version of Gromov hyperbolicity for differentiable and efficient computation. The method is evaluated on five unweighted graph datasets and two non-graph metric datasets, with results reported in terms of $\ell_\infty$ error.

**Questions:**

- Mean Average Precision (MAP) is a widely used evaluation metric in representation learning, especially for measuring tree and hyperbolic embedding quality. Could you report the MAP performance of your method compared to baselines?
- What are the δ-hyperbolicity or Gromov hyperbolicity values of the datasets used in your experiments? This would help contextualize how suitable these datasets are for tree-based representations.
- In Table 1, the LayeringTree method has missing entries (marked as “–”) for the ZEISEL and IBD datasets. Could you explain why these results are not reported?
- Are there any assumptions made about how data points are mapped to the tree? For instance, are points always assumed to be the leaves of the tree, or can they also include internal nodes?
- Is the resulting tree and its associated tree metric unique, given a distance matrix, or is the solution dependent on initialization or optimization variability?
- Beyond distance approximation, do you see your method being applicable to standard graph representation tasks such as node classification or link prediction? If so, how might your approach contribute to improved performance in those settings?

**Ethical Concerns:**

["NO or VERY MINOR ethics concerns only"]

**Final Justification:**

During the rebuttal and discussion period, the authors addressed my comments. I believe the work is important for tree metric learning and hierarchical representation communities.. The theoretical analysis is sound (Theorem 3.3), and should be appreciated. Therefore, I lean towards accepting the paper.

**Limitations:**

The author discuss the scalability issue for the proposed method in Section 5.

**Quality:**

3

**Strengths And Weaknesses:**

## Strength

Tree and tree metric learning is a specific form of hierarchical representation learning, which plays an important role in many machine learning tasks, especially given the prevalence of hierarchical structure in real-world data. The problem addressed in this paper is relevant to both the representation learning and hyperbolic geometry communities. The paper is clearly written and easy to follow. A key strength is the theoretical guarantee provided in Theorem 3.3, which shows that the proposed batched formulation asymptotically approximates Gromov hyperbolicity. I believe the theoretical contribution is solid and should be appreciated.

## Weakness
- In the formulation of the Minimum Distortion Tree Metric Approximation problem (line 35), the choice of using the $\|\|\cdot||\_\infty$  norm is not clearly justified. Many hyperbolic representation learning methods measure distortion using average distortion rather than worst-case. Since the proposed differentiable approximation of Gromov hyperbolicity is built upon the $\|\|\cdot||\_\infty$ norm, it would be important to explain why alternative norms (e.g., average distortion) are not suitable or less appropriate for this setting.
- The Gromov tree embedding referenced in Theorem 3.1 is not defined. Specifically, the construction of the mapping Φ is not described, which makes it difficult to follow the basis of the theorem.
- The optimization problem in Equation 3 and the Minimum Distortion Tree Metric Approximation problem (line 35) are based on the $\|\|\cdot||\_\infty$  norm, which motivates the use of a smooth relaxation due to the max and min operations. However, the final optimization objective in Equation 6 switches to using $\|\|\cdot||\_2\^2$, and the paper does not clarify this shift. If the initial motivation was centered around $\|\|\cdot||\_\infty$, the rationale for optimizing under a different norm should be made explicit.
- In Table 2, the $\ell_1$ average error is reported, but this metric is not formally defined in the paper. Moreover, based on this metric, the NJ method performs best and significantly outperforms the proposed method. Although NJ does not guarantee low worst-case distortion, the paper would benefit from a discussion or example where minimizing worst-case distortion (as targeted by the proposed method) is practically important.
- While the proposed approach focuses on tree metric learning, it is based on hyperbolicity, and the paper does not engage with related work in hyperbolic representation learning. Some prior methods also construct trees using hyperbolic space, and some prior hyperbolic representation learning methods can be trained in a differentiable manner. A discussion of these approaches and a comparison with relevant baselines would strengthen the paper and better position its contributions within the broader literature.

---

> ### Author Rebuttal · Authors · 2025-07-30
>
> ## Responses to weaknesses
>
> 1. Several works have investigated the metric nearness problem under the $\ell_\infty$ norm (e.g., [1,2,3,4]). Our choice is also motivated by Gromov’s theory of hyperbolicity, where the embedding theorem (Theorem 2.3 in our paper) is naturally formulated in terms of the $\ell_\infty$ norm. This makes $\ell_\infty$ the most consistent choice for linking distortion to guarantees linked to Gromov hyperbolicity. That said, we recognize the importance of alternative formulations in particular, Tree Metric Approximation under $\ell_p$ norms.
>    Our experiments suggest that using $\ell_2$ also leads to very good distorsion for $\ell_\infty$ norm.
>
> 2. Thank you for pointing this out. Appendix B provides full details; here is a concise summary.
> The Gromov embedding converts the input metric $d$ into
> $d\_G(x,y)=m-(x|y)\_w$, where $m=\max_{x} d(x,w)$.
> We then apply SLHC on $d_G$ to obtain an ultrametric
> $u_X$, and map back via $d_T(x,y)=m-u_X(x,y)$.
> The whole procedure costs $O(n^2)$ (see row 140). We will clarify this in the
> main text.
>
> 3. This is an excellent question. Our choice of the squared $\ell_2$ norm stems from its smoothness and differentiability, which make it more amenable to gradient-based optimization. Moreover, among common $p$-norms, $\lVert x\rVert\_2$ provides a good balance. It is closer to $\lVert x\rVert\_\infty$ than $\lVert x\rVert\_1$, as captured by the classical inequality in finite dimensional spaces:
> $$
> \lVert x\rVert_{\infty} \leq \lVert x\rVert_2 \leq \lVert x\rVert_1,
> $$
> This inequality highlights that minimizing the $\ell_2$ norm offers no direct control over the $\ell_1$ norm, which can result, during optimization, in lower $\ell_\infty$ distortion but higher $\ell_1$ distortion.
>
>    Importantly, Theorem 3.1 is derived from the classical Gromov tree approximation theorem (Theorem 2.3), which is fundamentally stated in terms of the $\ell_\infty$ norm. Consequently, the worst-case distortion guarantee in Theorem 3.1 holds only when distortion is measured using the $\ell_\infty$ norm. An interesting direction for future work would be to explore whether variants of the Gromov tree approximation theorem could be formulated under $\ell_p$ norms.
>
>    It is worth noting that, in principle, we could have used a smoothed approximation of the $\ell_\infty$ norm using the same log-sum-exp technique employed for the Gromov hyperbolicity. However, in our experiments, this alternative resulted in poorer optimization behavior and less stable outcomes. This observation further supports our use of the squared $\ell_2$ norm.
>
> 4. Thank you for raising this important point. We will add a formal definition of mean $\ell_1$ distortion (the average absolute error over all pairwise distances between the original and tree metrics). Although NJ performs well on this average error, it offers no control over the worst‑case distortion. Our method explicitly targets the $\ell_\infty$ norm, crucial when
>
>    - a single long edge can reshape a dendrogram,
>
>    - a bad estimate makes a network route unacceptably long, or
>
>    - one large error dominates a hierarchy’s visual interpretation.
>
>    We see our approach as complementary to NJ and other heuristics: while they perform well on average, they do not optimize or provide guarantees on the worst-case error.
>
> 5. We appreciate this observation and agree that our work could benefit from deeper engagement with the literature on hyperbolic representation learning. However, let us point **major differences between our work and hyperbolic embedding approaches**. Our method is rooted in Gromov hyperbolicity, a geometric property of metric spaces, and focuses on producing tree metrics. Related works in hyperbolic embeddings (e.g., Nickel & Kiela, 2017; Chamberlain et al., 2017) also aim to capture hierarchical structure, albeit from a different modeling perspective, generally based on contrastive losses that preserve relationships. In contrast, we operate directly on pairwise distances and focus on learning a surrogate metric that can be explicitly embedded into a tree with worst-case distortion guarantees. This metric-centric view differs from position/relation-based hyperbolic embeddings, but we see them as highly complementary. As we have noted in earlier responses, a promising direction for future work would be to optimize over point embeddings in hyperbolic space instead of full distance matrices. This could eliminate the need for explicit metric projection (via Floyd–Warshall) and would allow our method to tap into the rich literature and tools developed in the hyperbolic representation learning community.
>
> [1] "The metric nearness problem." SIAM Journal on Matrix Analysis and Applications, 2008.
>
> [2] "Metric nearness made practical." AAAI, 2023.
>
> [3] "Improved Error Bounds for Tree Representations of Metric Spaces." NIPS, 2016.
>
> [4] "UltraTWD: Optimizing Ultrametric Trees for Tree-Wasserstein Distance." ICML, 2025.
>
>
> ## Responses to questions
>
> 1. We thank the reviewer for this question, which echoes our previous answer on the relations with hyperbolic embedding approaches. In this specific context, where one aims at embedding graphs or trees in hyperbolic spaces, the MAP is a meaningful metric which measures how the relationships between neighbors is locally preserved through the embedding in a specific target space. **Recalling our objective is to find a tree which metric preserves at best a given input arbitrary metric**, we argue that the MAP does not necessarily measure the efficiency of our method nor the competitors for the following reasons:
>    - MAP does not measure how distances are preserved through the embedding (e.g. scaling uniformly the distances would yield the same MAP); It is only relevant on the preservation of local neighboring relations, and one could have simultaneously an arbitrary bad distortion with a perfect MAP score;
>    - An optimal tree, in the sense of our problem, can have a significantly different neighboring structure than an original graph (e.g. embedding a grid into a tree, which is an extreme case, illustrated in Figure 5 in the appendices);
>    - The embedding space should be the same for all the methods for the MAP score to be meaningful. Here, our methods and the competitors ones are embedding a metric into different tree structures (i.e. different spaces). As such, we can not directly compare the scores, as the target embedding spaces are different.
>
>    For those reasons, we decided to not compute the MAP score as demanded by the reviewer.  Nonetheless, if the reviewer still believes it can be a relevant metric to compare the methods, or at least give some insights, we will be happy to provide those scores.
>
> 2. We computed the hyperbolicity using the implementation provided in the SageMath library.
>
>    | Dataset  | Gromov hyperbolicity |
>    | -------- | ------ |
>    | C-ELEGAN | 1.5    |
>    | CS PHD   | 6.5    |
>    | CORA     | 4     |
>    | AIRPORT  | 1     |
>    | WIKI     | 2.5       |
>    | ZEISEL   |      0.19  |
>    | IBD      |  0.41      |
>
> 3. Thank you for the question. As briefly noted in line 287, the LayeringTree method is designed specifically for unweighted graphs and is not applicable to general metric datasets, such as ZEISEL and IBD. These datasets are based on pairwise dissimilarities derived from high-dimensional feature spaces and do not correspond to unweighted graphs with unit-length edges. As a result, LayeringTree cannot be applied in those settings, which explains the missing entries in Table 1.
>
> 4. We thank the reviewer for their question. There is indeed no assumption imposed on how data points are mapped to the tree. As illustrated in the visualizations provided in Figure 5, this mapping varies across methods:
>    - For example, HCC and NJ map all data points to the leaves of the constructed tree, which is standard for agglomerative methods.
>
>    - In contrast, our "Gromov embedding" approach allows data points to be mapped to internal nodes when this better captures the underlying geometry of the original space under the tree metric constraint (see Figure 6 for an example).
>
> 5. Empirically, we observe that our method produces consistent results across multiple runs, with only minor variations due to batch sampling. These variations have a limited impact on the final distortion, suggesting that our optimization is relatively stable in practice.
>
> 6. This is an excellent question. While our primary focus is on distance approximation, we believe our method has strong potential for standard graph‑representation tasks.
>
>    Our approach yields a surrogate distance metric that captures global and hierarchical structure in a differentiable, geometry‑aware manner. This enriched structural signal could be leveraged to produce high‑quality node embeddings, for instance, by embedding our optimized metric into hyperbolic spaces, or by providing it as input to downstream neural models.
>
>    In node classification, the tree‑like structure induced by our method may reflect latent class hierarchies or community structure. This intuition is supported by our experiments on stochastic block models (SBMs), where DELTAZERO helps to accurately recovers the underlying communities.
>
>    Furthermore, our method can be integrated as a pre‑processing step or regularizer in graph neural networks (GNNs), providing additional structural bias. This is especially relevant for addressing the well‑known over‑squashing phenomenon. Since Gromov hyperbolicity is tied to negative sectional curvature and over‑squashing has been linked to negative Ollivier–Ricci curvature [1], our framework offers a principled avenue to investigate, and potentially mitigate, such effects.
>
>    Exploring these directions is part of our ongoing and future work.
>
> [1] "Revisiting Over-smoothing and Over-squashing Using Ollivier-Ricci Curvature." ICML 2023.

---

> ### Comment · Reviewer_XSQu · 2025-08-02
>
> I would like to thank the authors for their detailed response. I’ve read both your rebuttal and the comments from the other reviewers, and I appreciate that most of my concerns have been addressed. One point I’m still curious about is how some of the clarifications you provided in the rebuttal will be reflected in the revision. For instance:
> - Will the discussion around the shift from $\|\| \cdot\|\|_\infty$ to $\|\| \cdot \|\|_2$ be included to help explain this?
> - Will there be a brief note, perhaps in the main text or appendix, on why MAP wasn’t considered, and how the goals of your method relate to those of hyperbolic representation learning more generally? I believe that including a short comparison or discussion, similar to what TreeRep did, would strengthen the paper
> - Will the hyperbolicity of the used datasets be included?
> - Will the discussion on over-squashing as a future direction be included as well?
>
> Clarifying how these will be incorporated would be very helpful.

---

> > ### Author Response · Authors · 2025-08-02
> >
> > We thank the reviewer for their thoughtful follow-up and suggestions. We will incorporate the clarifications as follows:
> > - We agree this clarification would benefit readers and will include a brief discussion in Section 3.2 to make the transition and motivation more explicit.
> > - We plan to add a better positioning with respect to hyperbolic methods in the introduction, alongside the formal statement of the Minimum Distortion Tree Metric Approximation problem and related methods. This will help contextualize our objectives and clarify the distinction between our approach and hyperbolic representations. In the experiments, we will also briefly explain why MAP is not relevant for evaluating the methods performance.
> > - We will include the Gromov hyperbolicity values of the datasets directly in Table 1, alongside existing statistics like distortion, number of nodes, and diameter.
> > - We briefly mentioned this direction on line 329. We will expand this with an additional sentence and a reference to the article cited in the rebuttal to better highlight its relevance as a future extension.
> >
> > We appreciate the reviewer’s suggestion to integrate these clarifications, which we believe will significantly strengthen the paper.

---

> > > ### Comment · Reviewer_XSQu · 2025-08-05
> > >
> > > I would like to thank the author for their effort to address my comments. I look forward to reading the revision.

---

### Official Review · Reviewer_FcsZ · 2025-07-03

**Clarity:** 4
**Significance:** 4
**Originality:** 4
**Rating:** 5
**Confidence:** 5

**Summary:**

This paper is motivated by the problem of finding a tree metric space embedding of a given arbitrary finite metric space that preserves the original pairwise distances. To this end, the authors propose an algorithm that consists of two major steps: 1) finding a surrogate finite metric space that best preserves the original distance and meanwhile has as close to a tree structure as possible; 2) embedding this surrogate into an actual tree using the existing Gromov’s tree embedding. The main contribution is in the first step. "preserving the original distances" is enforced by $l^\infty$ norm (in implementation $l^2$ norm is used for smoothness) and closeness to a tree is measured by Gromov hyperbolicity, which is not smooth by the min and max function in the definition. The authors formulate this problem as a constrained optimization problem with an objective as a balanced sum of these two loss components, where the Gromov hyperbolicity is replaced by a smooth version using the log-sum-exp operator and minibatch to alleviate the computation burden. Then the loss is differentiable and the optimization can be solved using projected gradient descent. The experiments show the benefit of this method in hierarchical clustering, and the quality of the embedding itself.

**Questions:**

1. In the batch sampling, the author mentioned the K subsets need to be independent, can you clarify what the independence here means? In Algorithm 1, the sampling seems just to be random, how is the independence enforced?
2. Can the authors explain what the blue vertices in the optimized tree of DeltaZero denote? Based on my understanding, the method preserves the number of points as the original input. I wonder why it would give rise to additional nodes.
3. Can the authors provide more discussion on why as all equivalent norms, minimizing the $l^2$ norm would yield better $l^\infty$ distortion but worse $l^1$ distortion? Also, how does using $l^2$ norm in implementations affect Theorem 3.1, if not, why don't you use $l^2$ norm in Equation (3) from the start?

**Ethical Concerns:**

["NO or VERY MINOR ethics concerns only"]

**Limitations:**

yes

**Quality:**

4

**Strengths And Weaknesses:**

Strengths:
1. This paper is well motivated. The embedding of an arbitrary metric into a tree is an interesting problem that is important in revealing the hierarchical structures of data, enabling fast tree algorithms, etc.
2. It reads very well, and I love that the authors care to give intuitive explanations to definitions, algorithms.
3. The smoothed version of Gromov hyperbolicity is novel.
4. Theoretical bounds are given for each approximation step of the Gromov hyperbolicity. The authors also provide critical analysis of the bounds, e.g. the restrictiveness of the assumptions.
5. Experiments show significantly improved accuracy of hierarchical clustering in synthetic data, and superior quality of the tree embedding of the proposed method over baselines. Sensitivity analysis is provided for better understanding of the parameters.


Weaknesses:
1. Although the optimization scheme may provide better worst-case bounds that give theoretical guarantees, in real-world applications the average-case bound is more relevant, and as shown in Table 2, the proposed method is not doing well.
2. Even after applying the batched approximation to reduce the computation complexity, the proposed algorithm takes ten to a hundred times longer than the baselines (Table 3). This hinders the application in large-scale data.
3. It would be better if the authors could show the applications such as hierarchical clustering in real datasets. But the experiments are sufficient in their current form.

---

> ### Author Rebuttal · Authors · 2025-07-30
>
> ## Reponses to weaknesses
>
> >Even after applying the batched approximation to reduce the computation complexity, the proposed algorithm takes ten to a hundred times longer than the baselines (Table 3). This hinders the application in large-scale data.
>
> You are absolutely right: our method has a complexity of $O(n^3)$, which makes it impractical for very large graphs. We would like to comment on few aspects:
> - We operate in the space of $n \times n$ distance matrices, meaning the per-step projection, though cubic, is not disproportionate relative to the size of the object being optimized ($n^2$ entries).
> - Several methods have been proposed to solve metric nearness (e.g. [1,2,3]), often relying on iterative procedures. While these methods may have lower theoretical complexity than Floyd–Warshall, they tend to be slower in practice. This is because Floyd–Warshall, despite its $O(n^3)$ complexity, can be efficiently parallelized on GPUs, offering significant practical speed-ups over iterative alternatives.
> - While being more computationally demanding than all competitors, it outperform baseline methods in terms of worst-case distortion, which is the central objective of our framework. It then provide a deliberate trade-off between computational cost and quality of approximation.
> - Many well-established methods in machine learning have similar complexities. It is nevertheless true that we still have room for improvement regarding the implementation of DeltaZero to make it run faster.
> - Lastly, to further improve scalability, a promising and natural direction we are exploring is to optimize over point embeddings in hyperbolic space instead of distance matrices. This would enforce metric validity by construction and eliminate the need for explicit projection steps, leveraging the efficiency of geometric optimization in continuous spaces.
>
> We firmly believe that this paper would constitute the first necessary cornerstone before exploring variants and optimized implementations for algorithms based on the idea of optimizing hyperbolicity.
>
> >It would be better if the authors could show the applications such as hierarchical clustering in real datasets. But the experiments are sufficient in their current form.
>
> Thank you for the suggestion. Exploring applications of our method on real-world hierarchical clustering tasks is indeed a direction we intend to pursue in future work. While our current focus was on controlled synthetic settings to clearly demonstrate the benefits of our optimization strategy, we believe that extending the method to practical use cases, such as biological taxonomies, document hierarchies, or social networks, is both promising and feasible. We see this as a natural continuation of the present work.
>
> [1] "The metric nearness problem." SIAM Journal on Matrix Analysis and Applications, 2008.
>
> [2] "Metric nearness made practical." AAAI, 2023.
>
> [3] "If it ain't broke, don't fix it: Sparse metric repair". IEEE, 2017.
>
> ## Responses to questions
>
> >In the batch sampling, the author mentioned the K subsets need to be independent, can you clarify what the independence here means? In Algorithm 1, the sampling seems just to be random, how is the independence enforced?
>
> In our setting, “independent” refers to the way batches are sampled during optimization:
>
> - Across batches: Each batch is sampled independently from the full dataset, meaning there is no dependence in how one batch affects the selection of the next.
> - Within a batch: Points are sampled without replacement, so each batch contains unique points (no duplicates within a batch).
> - Between batches: There is no enforced disjointness across batches, some overlap may occur, but the sampling process itself is statistically independent.
>
> We will clarify this in the final version to avoid ambiguity. Thank you for pointing this out.
>
> >Can the authors explain what the blue vertices in the optimized tree of DeltaZero denote? Based on my understanding, the method preserves the number of points as the original input. I wonder why it would give rise to additional nodes.
>
> The blue nodes represent additional points introduced by the Gromov embedding at the end of our optimization procedure. Although our method optimizes a metric over the original set of $n$ points, the final tree produced by the Gromov embedding of the optimized distance may include extra internal nodes in order to faithfully approximate the pairwise distances. This is a standard feature of Gromov’s construction: to embed a triplet of points $(x, y, w)$ into a tree, the algorithm may introduce an intermediate node $s$ located at distance $(x|y)_w$ from the base point $w$. This construction is illustrated in Figure 6 of the paper, where a three-point configuration is embedded into a tree with four nodes.
>
> >Can the authors provide more discussion on why, although all norms are equivalent in finite dimension, minimizing the $\ell\_2$ norm would yield better $\ell\_\infty$ distortion but worse $\ell\_1$ distortion?
>
> This is an excellent question. Our choice of the squared $\ell_2$ norm stems from its smoothness and differentiability, which make it more amenable to gradient-based optimization. Moreover, among common $p$-norms, $\lVert x\rVert\_2$ provides a good balance. It is closer to $\lVert x\rVert\_\infty$ than $\lVert x\rVert\_1$, as captured by the classical inequality in finite dimensional spaces:
> $$
> \lVert x\rVert\_{\infty} \leq \lVert x\rVert\_2 \leq \lVert x\rVert\_1,
> $$
> This inequality highlights that minimizing the $\ell_2$ norm offers no direct control over the $\ell\_1$ norm, which can result, during optimization, in lower $\ell\_\infty$ distortion but higher $\ell_1$ distortion.
>
> >Also, how does using the $\ell_2$ norm in implementations affect Theorem 3.1? If it does not, why don't you use the $\ell_2$ norm in Equation (3) from the start?
>
> Importantly, Theorem 3.1 is derived from the classical Gromov tree approximation theorem (Theorem 2.3), which is fundamentally stated in terms of the $\ell_\infty$ norm. Consequently, the worst-case distortion guarantee in Theorem 3.1 holds only when distortion is measured using the $\ell_\infty$ norm. An interesting direction for future work would be to explore whether variants of the Gromov tree approximation theorem could be formulated under $\ell_p$ norms.
>
> It is worth noting that, in principle, we could have used a smoothed approximation of the $\ell_\infty$ norm using the same log-sum-exp technique employed for the Gromov hyperbolicity. However, in our experiments, this alternative resulted in poorer optimization behavior and less stable outcomes. This observation further supports our use of the squared $\ell_2$ norm.

---

> > ### Comment · Reviewer_FcsZ · 2025-08-05
> >
> > I sincerely thank the authors for the clear and detailed responses to my concerns. Several notes from our discussion:
> >
> >
> > * I agree with the authors' comment about the computational cost of the proposed method, i.e., relative cost to input sizes, parallelizability without the need for iteration (as in baselines), better trade-off and potential for further optimization. I believe these important discussions should be incorporated into the revisions, including details on how it could be parallelized.
> > * Independence in batching sampling, additional vertices in the Gromov embedding step are well clarified.
> > * $l_2$ norm is a good choice in practice for its smoothness. By $ \lVert x\rVert_{\infty} \leq \lVert x\rVert_2 \leq \lVert x\rVert_1$, minimizing $l_2$ norm will yield less $l_\infty$ norm as used in Theorem 2.3, but not necessarily for $l_1$. A brief discussion of this choice would help clarify the narrative in the revision.
> > * Interesting future directions include optimizing the computation time and applications of our method on real-world hierarchical clustering tasks. I believe this paper has strong potential for significant impact.
> >
> > I have carefully read both the main paper and the appendix, and I believe the authors have sufficiently elaborated on the proposed method from both theoretical and experimental perspectives. With the minor revision mentioned above, this paper has the potential to make a strong contribution to the community.

---

> ### Author Response · Authors · 2025-08-06
>
> We thank the reviewer for their thoughtful suggestions and encouraging feedback. In response to all the points raised during the rebuttal, we plan to incorporate the following improvements into the revised version of the paper:
> - Computational cost: We will expand the discussion on computational complexity (around line 255), and in the Conclusion and Discussion section, we will emphasize the method’s potential for further optimization.
> - Batch sampling independence: We will add a clarifying sentence in the relevant section (line 193) to explain what we mean by independence in the batch sampling procedure.
> - Choice of norm in the loss: In Section 3.2, we will include a brief explanation of the transition from the $\ell_\infty$ norm (used in the theoretical bound of Theorem 2.3) to the $\ell_2$ norm (used in practice), emphasizing the trade-off between smoothness and approximation tightness, as discussed in our rebuttal.
> - Positioning relative to hyperbolic methods: We will strengthen the introduction by providing a clearer positioning of our approach with respect to hyperbolic embedding methods, which will help contextualize our contribution and differentiate it from hyperbolic representations.
> - Experimental clarifications: In the experiments section, we will briefly explain why Mean Average Precision (MAP) is not appropriate for evaluating our method’s performance. Additionally, we will include Gromov hyperbolicity values for all datasets directly in Table 1, alongside existing statistics such as distortion, number of nodes, and diameter.
> - Future directions: Finally, we will expand the Conclusion and Discussion section to highlight potential future applications of our method, especially in real-world hierarchical clustering tasks and possibly for other graph machine learning problems such as over-squashing in GNNs.
>
> We are grateful for the reviewer’s comments, which we believe will significantly strengthen the final version of the paper.

---

### Official Review · Reviewer_LGFS · 2025-07-03

**Clarity:** 3
**Significance:** 2
**Originality:** 2
**Rating:** 4
**Confidence:** 3

**Summary:**

The paper addresses the minimum-distortion tree metric approximation problem: given a finite arbitrary metric space $(X, d_X)$, the goal is to approximate it via a tree metric $d_T$ (with embedding $\Phi : X \to T$) that minimizes the worst-case distortion
\[
\max_{x,y} \bigl| d_X(x,y) - d_T\bigl(\Phi(x), \Phi(y)\bigr)\bigr|.
\]
This is motivated by the fact that tree metrics capture hierarchical structure, and any deviation from a tree metric can be measured by Gromov’s $\delta$-hyperbolicity (which is zero if and only if the metric is tree-like).

In this paper, the authors propose DELTAZERO, a differentiable optimization approach that ensures theoretical distortion guarantees over prior methods (for example, Neighbor-Joining, TreeRep, LayeringTree). It is a two-step process that jointly adjusts the metric to remain close to the original while enforcing small hyperbolicity (the trade-off between tree-likeness and fidelity to the original metric is governed by a regularization parameter $\mu$). In the first step, they introduce a surrogate metric space $\left(X', d_{X'}\right)$ and solve the optimization problem:
\[
\min_{d_{X'} \le d_X} \quad \mu \,\|d_X - d_{X'}\|_\infty \;+\; \delta\left( d_{X'} \right).
\]
To make this solvable by gradient-based methods, the authors replace the max/min definitions of $\delta$ with smooth log-sum-exp approximations and further approximate global hyperbolicity via batch sampling of point subsets. In practice, they optimize the distance matrix $D$ (subject to metric constraints) by projected gradient descent: each step uses ADAM, followed by projection onto the space of valid metrics via Floyd–Warshall. Finally, in the last step, the optimized metric $D^*$ is converted to a tree metric (for example, via a single-linkage hierarchical clustering procedure).

The paper is generally well-written and organized. Formal definitions are provided, which helps orient the reader. However, the content is mathematically heavy, and a reader may need a background in hyperbolic metrics to follow all the details.

The paper addresses a relevant problem important in several fields such as hierarchical clustering, phylogenetics, and network modeling. The approach of linking Gromov hyperbolicity to differentiable optimization is new and indicates high potential significance. Provided the experimental results hold up, the contribution has clear practical value.

The submission contains some original ideas as well as effective use of established techniques, combining them to achieve the results.

**Questions:**

1) Please explain the choice of parameters.


2) How do you ensure the $\mu\geq 1/(2\log(n-2))$ criterion holds?


3) How do you tackle the non-convexity of Gromov hyperbolicity $\delta_D$?

**Ethical Concerns:**

["NO or VERY MINOR ethics concerns only"]

**Final Justification:**

I am not convinced about the practicality of their approach and will therefore stick to my original score.

**Limitations:**

The submission does not specify the scheme of choosing optimal parameters, and how the non-convexity is tackled is not clear. Also, it doesn’t claim to perform well in general settings.

**Paper Formatting Concerns:**

None.

**Quality:**

3

**Strengths And Weaknesses:**

The submission seems to be technically sound overall.  The definitions and background are clearly stated, and proofs are deferred to appendices, indicating mathematical care. However, I have a few questions regarding some of the claims made.


Strengths:

1) The paper introduces a fresh perspective by combining Gromov hyperbolicity with differentiable optimization. Formulating the tree-approximation as a smooth optimization is innovative. The use of a log-sum-exp surrogate for hyperbolicity to make it differentiable for enabling gradient descent is an elegant idea. Unlike previous heuristics, this approach provides a framework that directly ties the distortion objective to a geometric measure (hyperbolicity).


2) The experimental results (as provided by the author(s) substantiate the claims. DELTAZERO consistently outperforms the standard algorithms currently in practice.


3) It does have some diverse and important future work potential.


Weakness:

1) Proposition 3.4 states that $\delta(D)$ is piecewise-affine but non-convex, so the smoothed surrogate remains non-convex. As a result, gradient descent may get stuck in local minima. The paper does not discuss initialization or how sensitive the results are to local optima. It uses ADAM and early stopping, but there is no guarantee the global optimum of the metric loss is found.


2) The method introduces several hyperparameters (batch size $m$, number of batches $K$, smoothing $\lambda$, tradeoff $\mu$, learning rate, etc.) Tuning these is nontrivial. The paper uses grid-search to find the best settings for each dataset. It is unclear how performance goes down if these are not optimally tuned. This reliance on manual hyperparameter search may lead to some limitations. Additionally, the condition in Theorem 3.1 for improved distortion ($\mu\geq 1/(2\log(n-2))$) may not be satisfied in practice, and its role is not deeply discussed.


3) While the method is compared against several baselines, the discussion of limitations of existing methods is brief. For instance, it is claimed that NJ, TREEREP, etc., “lack theoretical guarantees”. which is true for worst-case distortion but somewhat overstated in practical terms. Also, some baselines (like HCCROOTEDTREEFIT) depend heavily on root choice, but no analysis is given on how robust DELTAZERO is to analogous choices (e.g. the choice of basepoint $w$ in the final embedding). The “classical GROMOV” method mentioned is not fully explained in the main text (only in an appendix), making it unclear whether comparisons are fully fair.


4) The paper states that the surrogate metric is “eventually embedded into an actual tree” and mentions single-linkage clustering. However, details on the final embedding step (and its complexity) are omitted.This part invokes a variant of Gromov’s algorithm, but this area is only briefly sketched.


5) Though the current best known bound for computing Gromov hyperbolicity (which is $n^{3.69}$) is beaten, still each projected gradient step requires an $O(n^3)$ Floyd–Warshall projection to enforce triangle inequalities. While mini-batching reduces the cost of hyperbolicity estimation, “scalability to very large graphs” is still constrained by the cubic projection cost.


6) In some cases the improvements are small. For example, on the CORA graph DELTAZERO’s worst-case distortion (7.59) is only slightly better than the second-best (7.76 by LayeringTree), a 2.3% gain. This suggests that when the input metric is already very hyperbolic (small diameter relative to $n$), room for improvement might be limited. Also, In grid-like graph cases DELTAZERO is outperformed by LAYERINGTREE. The variability across datasets hints that in some regimes  the benefit may not be substantial.

---

> ### Author Rebuttal · Authors · 2025-07-30
>
> ## Responses to weaknesses
>
> 1. You are absolutely right, since our optimization problem is non‑convex, we cannot guarantee convergence to a global minimum. As shown in the paper (Algorithm 1) we always initialize with the original distance matrix, which is a theoretically motivated choice. Specifically, Theorem 3.1 ensures that if we could successfully minimize the distortion and Gromov hyperbolicity, the resulting tree metric will have a better distortion bound than the one obtained by directly applying the Gromov embedding.
>
>    Moreover, we empirically observe that our method produces consistent results across multiple runs, with only minor variations due to batch sampling. It is striking to notice that, in Table 1 and among all competitors, we have most of the time the lowest deviations observed across run.
>
> 2. We thank the reviewer for highlighting this important aspect. In Appendix D.2 of our submission (Figure 10), we provide a sensitivity analysis of the hyperparameters on the C-ELEGAN dataset. This analysis examines the impact of varying each parameter, namely $\mu$, $\lambda$ and K, individually, while keeping the others fixed at the values found through grid search. Our findings show that DELTAZERO exhibits stable performance under moderate deviations from the optimal settings, alleviating the concern that the method is overly sensitive to precise hyperparameter tuning.
>
>     The parameter $\mu$ is a chosen parameter that balances fidelity to the original metric against tree-likeness. Theorem 3.1 provides a theoretical condition on $\mu$, namely, $\mu \geq \frac{1}{2} \log(n-2)$ , under which, in the ideal case (i.e., if we were able to minimize both the $\ell\_\infty$ norm and Gromov hyperbolicity), the resulting tree embedding would guarantee a lower worst-case distortion than the classical Gromov embedding.
>
>     In practice **we do not have to enforce this constraint** strictly, since our optimization is performed over smooth surrogates of both objectives. Instead, $\mu$ is treated as a tunable hyperparameter, and its value is selected through grid search to empirically optimize distortion. We found that values below the theoretical threshold can still perform well in practice due to the regularizing effect of the smooth approximation.
>
> 3. Thank you for these thoughtful remarks. We agree that the practical performance of heuristic methods like NJ and TreeRep is often strong, and our statement about their lack of theoretical guarantees could have been more precisely worded. What we intended to emphasize is that, to the best of our knowledge, **these methods do not offer provable worst-case distortion bounds** of the kind established in Theorem 3.1 of our paper.
> That said, we acknowledge their practical utility and will revise our discussion to more accurately reflect their strengths.
>
>     For DeltaZero, the choice of basepoint only comes into play **after the optimization procedure**: we first optimize the distance matrix, and then apply the Gromov embedding to the optimized metric using 100 randomly selected basepoints. The reported average distortion is computed by averaging over these 100 embeddings (row 292/293). For fairness, we use the same set of 100 roots across all methods that require a basepoint (e.g., DELTAZERO, HCCRootedTreeFit, and Classical Gromov). This ensures a consistent and unbiased comparison, and helps reduce variability due to basepoint selection. We propose to add a discussion about this point in the revised version of the manuscript. We will also include more details about the Gromov-embedding procedure.
>
> 4. Thank you for pointing this out. We provided a more complete description in Appendix B, but we are happy to clarify it here. The Gromov embedding algorithm transforms a metric space into a tree metric using Gromov products with respect to a fixed base point $w$. For each pair $(x, y)$, it computes the Gromov product:
>     $$(x|y)\_w = \frac{1}{2} \bigl[d(x, w) + d(y, w) - d(x, y)\bigr],$$
>     then defines a transformed metric:
>     $$d\_G(x, y) = m - (x|y)\_w,$$
>     where $m = \max_{x \in X} d(x, w)$. This transformation is designed to encode ultrametric structure. Next, it uses single-linkage hierarchical clustering (SLHC) on $d_G$ to obtain an ultrametric $u_X$. The final tree metric is then recovered by inverting the transformation: $$d_T(x, y) = m - u_X(x, y).$$
>     The complexity is $O(n^2)$ as mentioned in row 140. We will clarify this in the main text.
>
> 5. You are absolutely right: our method has a complexity of $O(n^3)$, which makes it impractical for very large graphs. We would like to comment on few aspects:
>     - We operate in the space of $n \times n$ distance matrices, meaning the per-step projection, though cubic, is not disproportionate relative to the size of the object being optimized ($n^2$ entries).
>     - Several methods have been proposed to solve metric nearness (e.g. [1,2,3]), often relying on iterative procedures. While these methods may have lower theoretical complexity than Floyd–Warshall, they tend to be slower in practice. This is because Floyd–Warshall, despite its $O(n^3)$ complexity, can be efficiently parallelized on GPUs, offering significant practical speed-ups over iterative alternatives.
>     - While being more computationally demanding than all competitors, it outperform baseline methods in terms of worst-case distortion, which is the central objective of our framework. It then provide a deliberate trade-off between computational cost and quality of approximation.
>     - Many well-established methods in machine learning have similar complexities. It is nevertheless true that we still have room for improvement regarding the implementation of DeltaZero to make it run faster.
>     - Lastly, to further improve scalability, a promising and natural direction we are exploring is to optimize over point embeddings in hyperbolic space instead of distance matrices. This would enforce metric validity by construction and eliminate the need for explicit projection steps, leveraging the efficiency of geometric optimization in continuous spaces.
>
>     We firmly believe that this paper would constitute the first necessary cornerstone before exploring variants and optimized implementations for algorithms based on the idea of optimizing hyperbolicity.
>
> 6. In our experiments on real datasets, we report the gains over the second-best competitor. Those gains are always positive, meaning that we are **consistently** better than the competitors. We concur that there are regimes where this gain might be limited, but using DeltaZero systematically proved to be benefitial.
>
> [1] "The metric nearness problem." SIAM Journal on Matrix Analysis and Applications, 2008.
>
> [2] "Metric nearness made practical." AAAI, 2023.
>
> [3] "If it ain't broke, don't fix it: Sparse metric repair". IEEE, 2017.
>
> ## Responses to questions
>
> >Please explain the choice of parameters.
>
> The parameters in our method fall into three main categories:
>
> - Optimization parameters (e.g., learning rate, number of epochs, batch size): These were chosen using standard practices. The learning rate was selected via grid search from a fixed set of candidates.
>
> - Smoothing parameters (e.g., $\lambda$ in the log-sum-exp approximation): These control the smoothness of the surrogate for the max/min operations in the Gromov hyperbolicity. We selected these via grid search to balance approximation accuracy and numerical stability.
>
> - Trade-off parameter $\mu$: This parameter balances fidelity to the original metric (measured via $\ell_\infty$ norm) and tree-likeness (measured via smooth Gromov hyperbolicity). Theoretical guidance from Theorem 3.1 suggests setting $\mu ≥ 1/2 \log(n - 2)$ for worst-case distortion guarantees, but in practice we treated μ as a tunable hyperparameter and selected it via grid search. This choice reflects the fact that we optimize smooth surrogates rather than the exact hard constraints.
>
>     To ensure fairness and robustness, we applied the same grid search strategy for all datasets and reported the best-performing configuration for each case. Additionally, we include a sensitivity analysis in Appendix D.2, which shows that performance is generally stable across a reasonable range of hyperparameter values.
>
> >How do you ensure $\mu ≥ 1/2 \log(n - 2)$ the criterion holds?
>
> We believe we addressed this point in our response above.
>
> >How do you tackle the non-convexity of Gromov hyperbolicity $\delta_D$ ?
>
> We believe we addressed this point in our response above.

---

> > ### Comment · Reviewer_LGFS · 2025-08-04
> >
> > I would like to thank the author(s) for the detailed response. However, I am still not convinced about the practicality of their approach and will therefore stick to my original score.

---

### Official Review · Reviewer_ndmd · 2025-07-10

**Clarity:** 4
**Significance:** 3
**Originality:** 3
**Rating:** 4
**Confidence:** 4

**Summary:**

This paper introduces DELTAZERO, a novel differentiable optimization framework for approximating arbitrary metric spaces with tree metrics by leveraging a smooth surrogate for Gromov's $\delta$-hyperbolicity. The method is evaluated on both synthetic and real-world datasets, demonstrating state-of-the-art performance in minimizing distortion.

**Questions:**

1. Why use $\ell_\infty$ norm instead of Frobenius norm $||D_X-D||_F^2$ like metric nearness problems [1,2]?

2. Are there any other tree embedding method can be combined instead of Gromov’s method?

3. Is it possible to directly use the four-point conditions as the optimization constraints?

[1] "The metric nearness problem." SIAM Journal on Matrix Analysis and Applications, 2008.

[2] "Metric nearness made practical." AAAI, 2023.

**Ethical Concerns:**

["NO or VERY MINOR ethics concerns only"]

**Final Justification:**

**Final Justification for Score:** 4 (Borderline accept)​

**Resolved Issues:**

- The authors clarified their key innovation (optimizing $\delta$-hyperbolicity via gradient descent), resolving my initial concern.

- While still time-consuming, their justification (accuracy/runtime trade-off, GPU parallelization) and future plans (hyperbolic embeddings) make this acceptable.

- Early-stopping results show practical convergence; adding training curves would further strengthen the paper.

- The $\ell_\infty$/$\ell_2$ distinction and four-point condition discussion were helpful. Suggest including these in the paper/code.

**​​Recommendation​​:** Strengths outweigh limitations.

**Limitations:**

The primary limitations concern scalability and computational efficiency.

**Paper Formatting Concerns:**

None.

**Quality:**

4

**Strengths And Weaknesses:**

**Strengths**

1. The method addresses a fundamental problem in hierarchical representation learning.

2. The work provides theoretical foundations, including Proposition 3.2 bounding the approximation error, and Theorem 3.1 showing improved distortion guarantees.

---

**Weaknesses**

1. The method feels like it's combining existing techniques (projected gradient descent + Floyd-Warshall) rather than introducing something fundamentally new.

2. The $O(Km^4 + n^3)$ runtime seems pretty heavy - especially that $n^3$ part. For real-world use, we'd need to know how this scales with bigger graphs. Some scalability tests would really help show if this is practical.

3. Runtime performance is concerning - the method takes over 5600 seconds to process just 3005 nodes (Table 3 in Appendix D.1). This makes it impractical for real-world applications where we typically need to handle much larger graphs.

4. The paper doesn't discuss how quickly the method converges. It would be helpful to see how the solution quality improves with more iterations (T). Some analysis here would give users better intuition about tuning this parameter.

---

> ### Author Rebuttal · Authors · 2025-07-30
>
> ## Reponses to weaknesses
>
> >The method feels like it's combining existing techniques (projected gradient descent + Floyd-Warshall) rather than introducing something fundamentally new.
>
> We respectfully disagree with the reviewer on this point. While it is true that our optimization method builds on existing components, the core novelty of our work lies in optimizing the $\delta$-hyperbolicity through a soft approximation. In previous works, the $\delta$-hyperbolicity only appears to quantify how close a graph/dataset is to tree structure. Up to our knowledge, we are the first to propose optimizing it through gradient descent, and we expect that this idea has broader implications than the sole  Minimum Distortion Tree Metric Approximation problem which is the problem targeted in this paper.
>
> > Runtime performance is concerning - the method takes over 5600 seconds to process just 3005 nodes (Table 3 in Appendix D.1). This makes it impractical for real-world applications where we typically need to handle much larger graphs.
>
> You are absolutely right: our method has a complexity of $O(n^3)$, which makes it impractical for very large graphs. We would like to comment on few aspects:
> - We operate in the space of $n \times n$ distance matrices, meaning the per-step projection, though cubic, is not disproportionate relative to the size of the object being optimized ($n^2$ entries).
> - Several methods have been proposed to solve metric nearness (e.g. [1,2,3]), often relying on iterative procedures. While these methods may have lower theoretical complexity than Floyd–Warshall, they tend to be slower in practice. This is because Floyd–Warshall, despite its $O(n^3)$ complexity, can be efficiently parallelized on GPUs, offering significant practical speed-ups over iterative alternatives.
> - While being more computationally demanding than all competitors, it outperform baseline methods in terms of worst-case distortion, which is the central objective of our framework. It then provide a deliberate trade-off between computational cost and quality of approximation.
> - Many well-established methods in machine learning have similar complexities. It is nevertheless true that we still have room for improvement regarding the implementation of DeltaZero to make it run faster.
> - Lastly, to further improve scalability, a promising and natural direction we are exploring is to optimize over point embeddings in hyperbolic space instead of distance matrices. This would enforce metric validity by construction and eliminate the need for explicit projection steps, leveraging the efficiency of geometric optimization in continuous spaces.
>
> We firmly believe that this paper would constitute the first necessary cornerstone before exploring variants and optimized implementations for algorithms based on the idea of optimizing hyperbolicity.
>
> > The paper doesn't discuss how quickly the method converges. It would be helpful to see how the solution quality improves with more iterations (T). Some analysis here would give users better intuition about tuning this parameter.
>
> Thank you for your suggestion: in practive we observe that our method converges in a reasonable number of iterations. The following table gives for each dataset (with the hyper parameters choosed as in the paper i.e for the best obtained distortion) the number of epochs for which it has reached an early stopping criterion (defined as 50 consecutive gradient steps without improvement in the loss).
>
> These results show that convergence typically occurs well before the maximum of 1000 epochs, except in the WIKI dataset, suggesting that the optimization process is generally stable and effective. If the paper is accepted, we would be happy to include training loss curves in the supplementary material to illustrate empirical convergence across datasets.
> | **Dataset** | C-ELEGAN | CS-PHD | CORA | AIRPORT | WIKI | ZEISEL | IBD |
> | ------------------------------- | -------- | ------ | ---- | ------- | ----| ------ | --- |
> | **Epochs until Early Stopping** | 166      | 402    | 474  | 877     | 1000 | 456    | 122 |
>
>
> ## Responses to questions
>
> > Why use the $\ell_\infty$ norm instead of the Frobenius norm, like in metric nearness problems [1,2]?
>
> This is a relevant question. To clarify: we use $\ell_\infty$ only for theory and $\ell_2$ for practice.
> Precisely, our current definition of "Minimum Distortion Tree Metric Approximation" is grounded in Gromov’s theory, which uses  the $\ell_\infty$ norm, as it directly aligns with the worst-case distortion framework. That said, we recognize the value of alternative formulations, in particular under $\ell_p$ norms (e.g., $\ell_2$ or $\ell_1$) which relates to the metric nearness literature [1,2] as mentionned.
> In support of this, our experiments show that minimizing distortion in the $\ell_2$ norm also results in low distortion under $\ell_\infty$. We believe it would be interesting future work to leverage Gromov hyperbolicity to derive theoretical guarantees for tree metric approximations under general $\ell_p$ norms.
>
> >Are there any other tree embedding method can be combined instead of Gromov’s method?
>
> Thank you for this question, it's an important point that we can clarify. In principle, **any tree embedding method could be applied** to the metric obtained after our optimization step. We chose to apply the Gromov embedding because of its strong theoretical connection to Gromov hyperbolicity. In particular, Theorem 2.3 provides guarantees on the distortion based on the hyperbolicity constant, which aligns closely with the objective we optimize. This connection justifies our choice both theoretically and practically.
>
> >Is it possible to directly use the four-point conditions as the optimization constraints?
>
> This is an excellent remark. It is indeed possible to use a smooth version of the four-point condition as an optimization constraint. This requires implementing a smooth approximation of a sorting operation (see for instance [4]), since the four-point condition involves identifying the maximum among three sums of distances. In our work, we chose to formulate the constraint via the Gromov product instead, using the log-sum-exp function to approximate both the max and min. We had implemented both and they gave similar results.
>
> [1] "The metric nearness problem." SIAM Journal on Matrix Analysis and Applications, 2008.
>
> [2] "Metric nearness made practical." AAAI, 2023.
>
> [3] "If it ain't broke, don't fix it: Sparse metric repair". IEEE, 2017.
>
> [4] "Fast Differentiable Sorting and Ranking." ICML, 2020.

---

> > ### Comment · Reviewer_ndmd · 2025-08-09
> >
> > Thank you for your thorough clarification and additional results. My concerns have been fully addressed, and I'm happy to give a positive score of 4.
> >
> > I would encourage you to incorporate the discussions from your rebuttal into the revised version, particularly regarding computational complexity, convergence behavior, and the connection with metric nearness. Additionally, it would be valuable if you could include your implementation of the four-point condition in the open-source code (if possible), as I'm particularly interested in the algorithmic implementation and performance aspects.

---

### Note · Authors · 2025-08-13

We thank the reviewers for the constructive and insightful feedback on our work. We are confident that incorporating the suggested addition covering: computational cost, batch sampling independence, choice of norm in the loss, positioning relative to hyperbolic embedding methods, convergence behavior, connections with metric nearness, and further experimental details, will substantially strengthen the paper.

---

### Decision · Program_Chairs · 2025-09-17

**Decision:**

Accept (poster)

**Comment:**

This paper introduces DeltaZero, which optimizes deviations from a tree metric through a smooth differntiable surrogate of Gromov's delta-hyperbolicity. Experiments on several graph datasets shows that DeltaZero consistently improves over baseline embedding algorithms. After the review and discussion phase, all reviewers provide recommendations above the acceptance threshold (5, 4, 4, 4). The reviewers find the approach novel and well motivated, while the paper is also clearly presented. There were also multiple concerns, shared by (nearly) all reviewers. Notable are the concerns regarding computational complexity, convergence, and choice of norms (specifically, Linf). The authors provide a clear and elaborate rebuttal, agreeing on the computational complexity and motivating the choice of norms. The AC agrees with the reviewers that these issues remain an open question, especially when it comes to large-scale real-world settings. However, the paper has sufficient novelty and empirical impact to warrant acceptance. The AC follows the reviewer recommendations and votes for acceptance.